# Model-based fMRI reveals dissimilarity processes underlying base rate neglect

**Sean R O'Bryan[1]\*, Darrell A Worthy[2], Evan J Livesey[3], Tyler Davis[1]**

[1]Department of Psychological Sciences, Texas Tech University, Lubbock, United States; [2]Department of Psychology, Texas A&M University, College Station, United States; [3]School of Psychology, University of Sydney, Sydney, Australia

**Abstract** Extensive evidence suggests that people use base rate information inconsistently in decision making. A classic example is the inverse base rate effect (IBRE), whereby participants classify ambiguous stimuli sharing features of both common and rare categories as members of the rare category. Computational models of the IBRE have posited that it arises either from associative similarity-based mechanisms or from dissimilarity-based processes that may depend on higher-level inference. Here we develop a hybrid model, which posits that similarity- and dissimilarity-based evidence both contribute to the IBRE, and test it using functional magnetic resonance imaging data collected from human subjects completing an IBRE task. Consistent with our model, multivoxel pattern analysis reveals that activation patterns on ambiguous test trials contain information consistent with dissimilarity-based processing. Further, trial-by-trial activation in left rostrolateral prefrontal cortex tracks model-based predictions for dissimilarity-based processing, consistent with theories positing a role for higher-level symbolic processing in the IBRE.
DOI: https://doi.org/10.7554/eLife.36395.001

## Introduction

Does this patient have influenza or Ebola virus? Categorization is a fundamental process that underlies many important decisions. Categories, such as viruses, often have different relative frequencies or base rates. Influenza, for example, is very common and infects millions of people worldwide each year, whereas Ebola virus tends to have infection rates that are orders of magnitude lower.

One critical question is how people use such base rate information when making categorization decisions. Research so far has suggested that people tend to be, at best, inconsistent in their use of base rate information. In both realistic studies with medical professionals and artificial categorization tasks in the lab, when confronted with examples that share characteristics with both rare and common categories, people show a tendency to predict the rare category much more often than the base rates would suggest (*Tversky and Kahneman, 1974*; *Casscells et al., 1978*; *Bravata, 2000*). In an extreme case, known as the inverse base rate effect (IBRE), people may even predict rare categories as more likely than common ones (*Medin and Edelson, 1988*). For example, in an IBRE context, a patient presenting with cough (a characteristic feature of influenza) and unexplained bleeding (a characteristic feature of Ebola), may be more likely to be diagnosed with Ebola than influenza.

The mechanisms that lead to base rate neglect are currently undetermined at both the cognitive and neural levels. Computationally, according to influential work with similarity-based categorization models (*Medin and Edelson, 1988*; *Kruschke, 1996*, *Kruschke, 2001*), the IBRE arises from differential selective attention to features for common and rare categories. Specifically, participants learn to attend more strongly to features of rare categories, making ambiguous cases seem more similar to rare categories and thus more likely to be rare category members. In terms of the flu example, participants may attend more to the unexplained bleeding feature of the rarer Ebola virus category, and thus predict Ebola when confronted with a patient with both features.

**\*For correspondence:**
sean.r.obryan@ttu.edu

**Competing interests:** The authors declare that no competing interests exist.

**eLife digest** Is a patient with muscle aches, headache and fever more likely to have influenza or Ebola? Most people correctly choose 'influenza' because it is the more common of the two diseases. But what about someone with a cough and unexplained bleeding? Coughing is a symptom of influenza but not of Ebola. Unexplained bleeding is a symptom of Ebola but not of influenza. Faced with ambiguous symptoms such as these, many people would diagnose 'Ebola' despite knowing that influenza is more common. Indeed, when a situation shares characteristics with both a common and a rare category, we often tend to predict that it belongs to the rare group. This phenomenon is known as base rate neglect, but why does it occur?

One theory is that we pay more attention to features that belong to rare categories (such as unexplained bleeding) because they are distinctive and unusual. But another possibility is that we use our knowledge of the common category to rule out examples that do not conform to our expectations. Of the many cases of influenza that you have heard about or experienced, probably none of them featured unexplained bleeding.

To distinguish between these possibilities, O'Bryan et al. trained healthy volunteers on a categorization task that included ambiguous stimuli. The participants performed the task inside a brain scanner. O'Bryan et al. then programmed a computer to solve the same problems. The simulation either used similarity-based judgments (how similar is this to the rare category?), dissimilarity-based judgments (how dissimilar is this from the common category?), or both. The results suggested that when people show base rate neglect, they rely more on dissimilarity-based evidence than on similarity-based evidence. In other words, they focus more on how a test item differs from the common category. Consistent with this, whenever the volunteers chose the rare category, their brains were processing information about the common category. The imaging results also revealed that when the volunteers used dissimilarity-based evidence, they activated a brain region involved in abstract thinking and reasoning.

How people use information about likelihoods is relevant to all aspects of decision-making. Beyond helping us to understand how we assign items to categories, the work by O'Bryan et al. could also inform future research in areas such as learning and memory.

DOI: https://doi.org/10.7554/eLife.36395.002

Similarity-based category learning models have strong support in the neurobiological category learning literature. Model-based predictions for how similar items are to stored category representations have been shown to correlate with activation in the medial temporal lobes (MTL; *Davis et al., 2012a*, *Davis et al., 2012b*). Moreover, at a finer-grained level, multivoxel activation patterns in the MTL have been shown to contain information associated with higher-order similarity relationships between category members anticipated by similarity-based models (*Davis and Poldrack, 2014*), including those predicted by differences in selective attention (*Mack et al., 2016*). The dorsolateral prefrontal cortex (dlPFC) tends to track predictions of choice uncertainty from similarity-based models, whereas ventromedial PFC (vmPFC) tends to track estimates of high choice accuracy or model-confidence (*Davis et al., 2017*).

Despite the strong cognitive and neural evidence for similarity-based models, it remains an open question whether they provide a complete account of IBRE-like phenomena. One alternative proposition is that people's choice of rare categories when confronted with conflicting information may stem from reliance on dissimilarity processes, either solely, or in addition to similarity-based processes. According to theories that focus on dissimilarity-based processes, people build strong expectations of the common category; thus they view items containing features inconsistent with these expectations as more likely to be members of the rare category (*Juslin et al., 2001*; *Winman et al., 2005*). For example, a doctor may have seen thousands of cases of flu, none with unexplained bleeding, and thus rule out influenza and choose Ebola virus based on these expectations. In these cases, it is *dissimilarity* to members of the common (unchosen) category that drives choice, rather than the similarity to rare (chosen) category members per se.

Formal models positing dissimilarity processes have so far been explicitly dual-process oriented. For example, ELMO, a computational model that incorporates a choice elimination decision based

on dissimilarity, argues that such elimination depends on explicit reasoning processes that are separate from similarity-based processes that arise in other trials (*Juslin et al., 2001*). In the present study, we propose a new account based on a recently proposed dissimilarity-based extension of the generalized context model, the dissGCM (*Stewart and Morin, 2007*). This account uses the same basic similarity computations as standard similarity-based models (e.g. *Nosofsky, 1986*), but allows similarities and dissimilarities to stored exemplars to be used as evidence for a category. In terms of the above example, dissimilarity to influenza can be used as evidence for Ebola (and vice versa).

As specified computationally, the dissGCM is agnostic about whether using dissimilarity-based evidence constitutes a different cognitive or neurobiological mechanism from using similarity-based evidence. On one hand, the dissGCM has no fundamentally different computations from a basic similarity process; as detailed below, dissimilarity is a simple transformation of similarity. On the other hand, it is possible that dissimilarity processes require manipulation of similarity relationships between category representations in a more symbolic or abstract manner, as anticipated by previous dissimilarity theories. Given that the dissGCM makes separable estimates for the relative contributions of similarity- and dissimilarity-based evidence to choice in a given trial, these model predictions can be used to explicitly test whether employing dissimilarity-based evidence against unchosen categories engages brain regions beyond those associated with the use of similarity-based evidence, consistent with dual-process accounts of IBRE. Specifically, we can test whether regions that are known to be critical for using higher-level abstract rules track dissGCM's predicted trial-by-trial used of dissimilarity-based evidence, and whether these regions diverge from those typically found to track estimates of similarity-based evidence.

Higher-level cognitive control mechanisms are thought to depend on a hierarchy of abstraction in the lateral PFC along the rostral-caudal axis (*Badre and D'Esposito, 2007*; *Badre and D'Esposito, 2009*). At the apex of this hierarchy is the rostrolateral PFC (rlPFC), a region often implicated in tasks that require people to generalize across abstract, symbolic representations. For example, relational reasoning tasks such as Raven's progressive matrices and rule-based tasks involving abstract relations are thought to depend on left rlPFC (*Christoff et al., 2001*; *Bunge et al., 2005*, *Bunge et al., 2009*; *Davis et al., 2017*). In addition to its role in generalizing abstract, relational rules, we have recently found left rlPFC to be involved in rule evaluation and novel generalization processes for simpler feature-based rules in categorization tasks (*Paniukov and Davis, 2018*). In the present study, dissimilarity-based generalization to novel feature pairings may depend on rule evaluation processes in the rlPFC more so than simple similarity-based processing, if studies anticipating that dissimilarity-based processes depend more upon higher-level symbolic rules are correct (*Juslin et al., 2001*; *Winman et al., 2005*). Alternatively, pure similarity-based accounts suggest that generalization patterns in an IBRE task do not depend on the existence of a separate, higher-level mechanism (*Medin and Edelson, 1988*; *Kruschke, 1996*, *Kruschke, 2001*), and would thus expect a single neurobiological network associated with similarity-based processing to be engaged for choice across trials.

Here we test the dissGCM by incorporating its predictions into an analysis of fMRI data collected from participants completing a standard IBRE task (*Medin and Edelson, 1988*; *Kruschke, 1996*). We first examine whether multivoxel activation patterns elicited during conflicting trials in the IBRE task are consistent with participants activating information associated with the rare category, as predicted by pure similarity-based accounts, or activating information associated with (dissimilarity to) the common category, as predicted by the dissGCM. To this end, we use representational similarity analysis (RSA; *Kriegeskorte et al., 2008*) to decode which features of the stimuli are most strongly activated while participants are categorizing the conflicting items. This analysis is based on recent work in the broader memory literature establishing that it is possible to decode whether participants are retrieving particular object categories from memory based on their activation patterns in ventral temporal cortex (*Rissman and Wagner, 2012*; *Haxby et al., 2014*).

To facilitate the multivoxel analysis, here we use the real world visual categories faces, scenes, and objects as stimulus features. These visual categories have a well-defined representational topography across the cortex (*Haxby et al., 2001*; *Grill-Spector and Weiner, 2014*), allowing us to predict whether participants are differentially activating particular stimulus features (faces, scenes, or objects) by computing similarities between activation patterns elicited for the key IBRE trials and feature-specific patterns from an independent localizer scan. By crossing the visual stimulus features with our category structure (*Figure 1*), we create situations where a rare category is associated with

one feature type (e.g. a scene) and a common category is associated with another feature type (e.g. an object). The extent to which each type of information is active can then be compared to determine whether participants are representing stimulus features associated with the common or rare category on a trial, and thus answer whether their BOLD activation patterns are more consistent with pure similarity or dissGCM's combined dissimilarity and similarity processes. In this context, we anticipate that the multivoxel pattern analysis will index an interactive process between feature-based attention and memory retrieval: the dissGCM, pure similarity-based GCM, and previous dissimilarity-based inference models all predict that categorization decisions are driven by an attention-weighted memory process whereby a stimulus is compared with the contents of memory (*Nosofsky, 1986*; *Juslin et al., 2001*; *Stewart and Morin, 2007*). This prediction suggests that during categorization, the multivoxel patterns activated for a particular stimulus will reflect both direct perceptual processing and retrieval of information from memory. Because the dissGCM predicts greater contributions from the common, unchosen category during this retrieval process, we expect multivoxel patterns during ambiguous trials to reveal greater activation of information associated with the common, unchosen category.

In addition to our multivoxel analysis, we also test whether using dissimilarity-based evidence against unchosen categories may tap distinct brain regions, such as the rlPFC, beyond those involved with similarity-based computations. To this end, we take trial-by-trial predictions for how much similarity- and dissimilarity-based evidence contribute to the winning category and use these predictions as regressors in fMRI analysis. We anticipated that the MTL and vmPFC would be positively associated with similarity-based evidence, whereas dlPFC would be negatively associated with similarity-based evidence for the winning category. Contrastingly, we expected rlPFC to track estimates of dissimilarity-based evidence against alternative options.

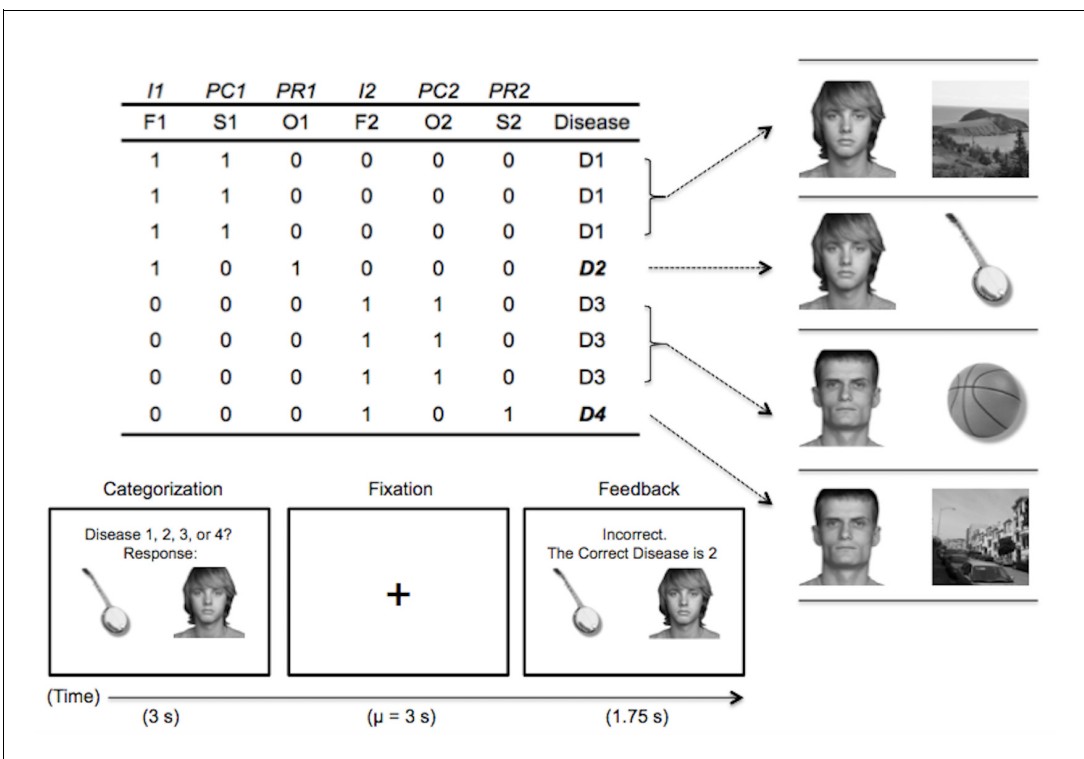

**Figure 1.** Abstract task design and an example trial. In the headings, I = imperfect predictor, PC = common perfect predictor, PR = rare perfect predictor. The second row refers to the visual category used for each stimulus feature: F = face, S = scene, O = object. Each following row corresponds to a learning trial, with a '1' indicating the presence of the feature and '0' indicating its absence.
DOI: https://doi.org/10.7554/eLife.36395.003

# Results

## Behavioral results and model fit

Learning curves over the 12 learning blocks for common and rare disease item pairs are shown in *Figure 2*. All subjects reached greater than 90% accuracy over the last four blocks (M = 98.1%, SD = 2.4%, range = 93.5–100%). Mean choice performance in the first block was above chance (25%) for both common (M = 63.6%) and rare (M = 43.2%) feature pairs. Consistent with previous IBRE studies, a linear mixed effects model revealed a significant block by trial type interaction ($F_{(1, 262)}=20.7$, $p<0.001$), suggesting that the common diseases were learned more quickly than the rare diseases. Paired *t*-tests revealed that participants were significantly more accurate on common compared with rare disease trials in the first ($t_{(21)}=2.26$, $p=0.034$), second ($t_{(21)}=2.85$, $p=0.010$), third ($t_{(21)}=2.72$, $p=0.013$), fourth ($t_{(21)}=2.46$, $p=0.023$), and 12th blocks ($t_{(21)}=2.23$, $p=0.037$).

For the test phase, participants were asked to categorize the original category exemplars in addition to a number of other novel feature combinations in the absence of feedback. We then fit the dissGCM to the group choice probabilities for each test item. The dissGCM is based on the original generalized context model (*Nosofsky, 1986*), but allows for dissimilarity to be used as evidence for a decision (*Stewart and Morin, 2007*). The model posits that people represent stimuli as points in a multidimensional feature space, and that categorization judgments are based on distances between probe stimuli and stored exemplars. As for the standard GCM, similarities to all exemplars of each category are summed into evidence for each category. However, in the dissGCM, evidence that an item is dissimilar to *other* categories is also used as evidence for a category. For example, evidence for Disease 1 includes not only an item's similarity to members of Disease 1, but also its dissimilarity to other diseases.

Choice probabilities and dissGCM-derived predictions for each of the test items are summarized in *Table 1*. Consistent with an inverse base rate effect, participants were numerically more likely to classify ambiguous test stimuli (combinations of rare and common features) as members of the

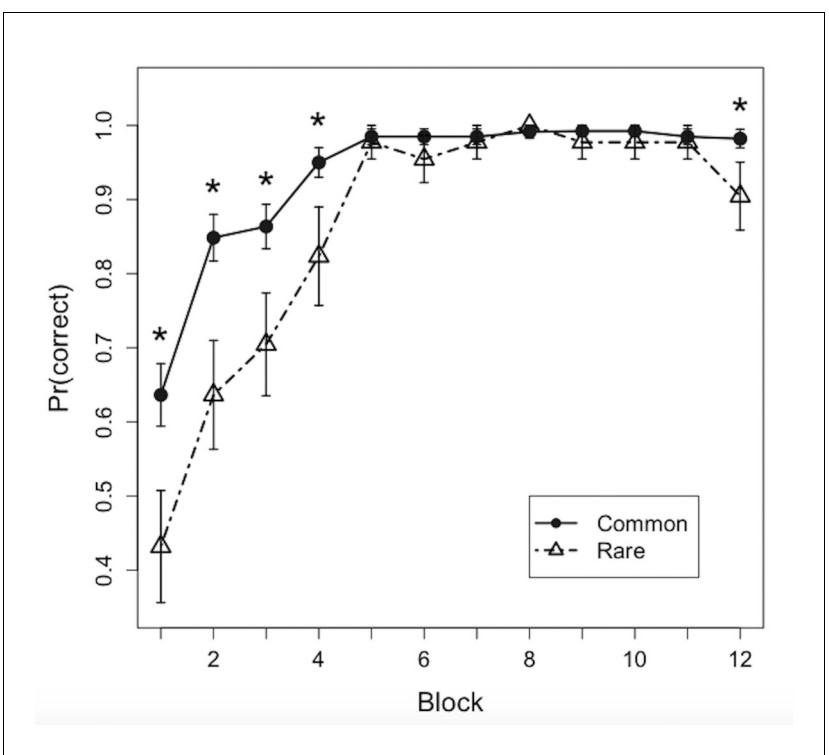

**Figure 2.** Learning curves. Points depict proportions correct for common and rare disease predictions over the 12 blocks of the training phase (mean ± SEM). *p < 0.05.
DOI: https://doi.org/10.7554/eLife.36395.004

relevant rare category (M = 49.8%) than the relevant common category (M = 43.5%) combined across object-scene, scene-scene, and object-object pairs. A one-sample t-test revealed that the percentage of rare responding on ambiguous trials was significantly higher than the 1/4 base rate for the rare category ($t$ (21)=8.11, p<0.001). Likewise, participants chose the rare category for ambiguous pairs significantly more often than for the imperfect predictors (faces: M = 30.0%), ($t$ (21)=3.85, p<0.001).

In addition to response probabilities, we tested whether reaction times differed on the ambiguous test trials depending on whether a rare or common response was made. On these trials of interest, a linear mixed effects model revealed that RTs were slower when participants made rare responses (M = 1.47 s) than common responses (M = 1.27 s), ($t$ (21)=10.48, p<0.001). The observation of slowed RTs on ambiguous trials receiving rare responses suggests that rare selections may be more cognitively demanding relative to common selections, consistent with previous dissimilarity-based theories of IBRE that posit a role of higher-level, inferential reasoning in base rate neglect.

## Multivoxel results

### Test phase

The primary goal of the multivoxel analysis was to decode, for the ambiguous stimuli, whether participants were activating information consistent with the common or rare category when they make the choice to classify the stimulus as rare. Specifically, for the bold italicized stimuli listed in *Table 1*, we tested whether participants' activation patterns were more similar to localizer activation patterns associated with scenes when a scene was the common feature (and object was rare) and more similar to those of objects when an object was the common feature (and scene was rare). Participants

---

**Table 1.** Observed and dissGCM-predicted response probabilities for the test phase.

The feature combinations presented at test are listed in the leftmost column: F = face, S = scene, O = object. In the headings, D1–D4 correspond to the four possible category responses (diseases). Bold, italicized values indicate results for the key ambiguous stimuli in which a scene was paired with an object.

| Test item | Behavior | | | | dissGCM | | | |
|---|---|---|---|---|---|---|---|---|
| | D1 | D2 | D3 | D4 | D1 | D2 | D3 | D4 |
| F1 + S1 | .972 | .018 | .008 | .003 | .971 | .014 | .008 | .008 |
| F1 + O1 | .031 | .962 | .008 | 0. | .063 | .901 | .018 | .018 |
| F2 + O2 | .005 | 0. | .987 | .008 | .008 | .008 | .972 | .013 |
| F2 + S2 | .023 | .008 | .069 | .901 | .018 | .018 | .059 | .905 |
| F1 | .667 | .295 | .023 | .015 | .667 | .283 | .025 | .025 |
| F2 | .107 | .038 | .550 | .305 | .027 | .027 | .640 | .307 |
| S1 | .848 | .061 | .008 | .083 | .955 | .015 | .015 | .015 |
| O1 | .008 | .908 | .069 | .015 | .040 | .880 | .040 | .040 |
| O2 | .008 | .069 | .908 | .015 | .012 | .012 | .965 | .012 |
| S2 | .023 | .053 | .008 | .916 | .032 | .032 | .032 | .904 |
| *S1 + O1* | *.414* | *.487* | *.073* | *.027* | *.419* | *.496* | *.042* | *.042* |
| *O2 + S2* | *.035* | *.047* | *.453* | *.465* | *.038* | *.038* | *.460* | *.464* |
| F1 + O2 | .131 | .238 | .631 | 0. | .156 | .174 | .658 | .013 |
| F1 + S2 | .264 | .062 | .008 | .667 | .178 | .237 | .023 | .563 |
| F2 + S1 | .608 | .031 | .138 | .223 | .657 | .013 | .155 | .175 |
| F2 + O1 | .008 | .674 | .302 | .016 | .024 | .567 | .170 | .239 |
| O1 + O2 | .008 | .514 | .475 | .004 | .040 | .444 | .477 | .040 |
| S1 + S2 | .397 | .065 | .011 | .527 | .400 | .040 | .040 | .520 |

DOI: https://doi.org/10.7554/eLife.36395.005

---

encountered 24 examples of these key object-scene pairings over the course test phase, and the choice patterns for each subject are detailed in *Supplementary file 1*.

The prediction that information associated with the common category should be more active on ambiguous trials is derived from the dissGCM. When examining how much each category's exemplars (common and rare) contributed to the rare response for ambiguous items, the model posits that rare choices are more probable because of the contribution that dissimilarity to the common category makes to the evidence for the rare category relative to the contribution of similarity to the rare category. Formally, similarity-based evidence is given as the summed attention-weighted similarity of a stimulus to the winning category (*Equation 5* in Materials and methods), and dissimilarity-based evidence is the summed attention-weighted dissimilarity to the non-winning categories (*Equation 6* in Materials and methods). For example, using these formulas, in the fitted version of the model, the proportion of the overall evidence for rare contributed by similarity to the rare category exemplar was nearly half the evidence contributed by dissimilarity to the common category exemplar (rare = 0.088; common = 0.153, in the dissGCM's attention weighted similarity units).

For this analysis, multivoxel pattern estimates were anatomically restricted to ROIs in ventral temporal cortex associated with each visual stimulus category (objects: left inferior posterior temporal gyrus; scenes: bilateral parahippocampal gyrus; and faces: right temporal occipital fusiform gyrus). Within these respective regions, subject-level ROIs were then functionally selected by creating 6 mm spheres around subjects' peak activation to each category during the localizer phase. Pattern estimates were found to discriminate between information associated with control items during test: pattern similarity estimates for objects were significantly greater on object-only trials (O1, O2, and O1+O2) than scene-only trials (S1, S2, and S1+S2), ($t$ (21)=2.83, p=0.010), and vice versa ($t$ (21) =5.60, p<0.001). Likewise, estimates for faces on face-only control trials were found to be significantly greater than on object-only trials ($t$ (21)=2.54, p=0.019), and scene-only trials ($t$ (21)=2.92, p=0.008).

Multivoxel pattern similarity results for the ambiguous test trials are depicted in *Figure 3*. Consistent with the dissGCM's predictions, a linear mixed effects model with BOLD pattern similarity as the outcome variable and categorical predictors for response, stimulus dimension (common vs. rare), and visual stimulus category (objects vs. scenes) revealed a significant interaction between response and pattern similarity to common and rare features, whereby participants tended to more strongly activate patterns associated with common features only when they made a rare response ($F$ (1, 42) =4.92, p=0.032). Specifically, when participants chose the rare category, their activation patterns were most similar to whichever visual stimulus category (scenes or objects) was associated with the common category ($t$ (21)=2.78, p=0.011). Interestingly, there was no significant difference between pattern similarity for rare and common features when participants made a common response ($t$ (21) =0.45, p=0.653). Visual stimulus category was not found to interact with response ($F$ (1, 72)=0.229, p=0.634), or whether an item was rare or common ($F$ (1, 72)=0.241, p=0.625), in the pattern similarity model, and thus the results in *Figure 3* are collapsed across objects and scenes. A depiction of the test phase results including means for each distinct item in the model can be found in *Figure 3— figure supplement 1*.

Because faces were off-screen for the key test trials and pattern similarity to the face dimension could represent information associated with either common or rare exemplars, no a priori predictions were made regarding pattern similarity to faces on ambiguous trials. However, a one-sample t-test revealed no significant differences in pattern similarity to the face dimension across responses ($t$ (21)=0.22, p=0.828). Mean pattern similarity to faces on the ambiguous test trials is depicted by the gray squares in *Figure 3*.

To summarize, our multivoxel findings for the test phase suggest that people more strongly activate information associated with common categories when engaging in base rate neglect, consistent with dissGCM's prediction that dissimilarity to the common category exemplar contributes more to rare decisions than similarity to the rare category exemplar. Although the model's evidence weighting predictions and our multivoxel results provide a reasonable account for why participants tend to choose rare categories for ambiguous stimuli, the dissGCM does not address the question of whether a separate mechanism or strategy may contribute to trials in which the common category is chosen. Like the dissGCM, previous dual-process theories of IBRE (*Juslin et al., 2001*; *Winman et al., 2005*) propose that rare responses tend to be a byproduct of dissimilarity-based

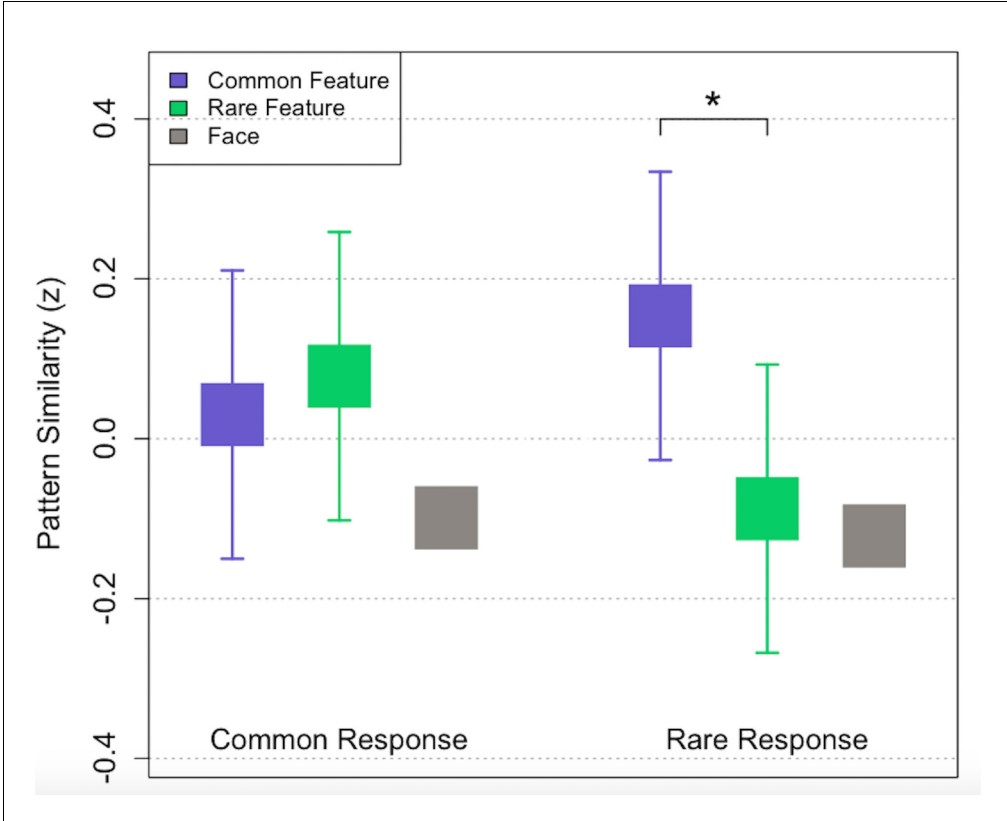

**Figure 3.** Multivoxel pattern similarity to common and rare stimulus features for ambiguous trials in which participants made common (left) and rare (right) responses (mean ± SEM). Purple squares correspond to the objects/scenes associated with the common category, while green squares correspond to the objects/scenes associated with the rare category in a given trial. Gray squares depict mean pattern similarity to the non-present face dimension for both response types. *p < 0.05. No error bars are included for the gray bars because face dimensions were not included in the overall mixed effects model.

DOI: https://doi.org/10.7554/eLife.36395.006

The following figure supplement is available for figure 3:

**Figure supplement 1.** Multivoxel pattern similarity to common and rare objects, scenes, and faces for ambiguous trials in which participants made common (left) and rare (right) responses (mean ± SEM).

DOI: https://doi.org/10.7554/eLife.36395.007

processing, but hypothesize that common responses are more likely a result of a 'strategic guessing' strategy that is engaged when a probe fails to elicit a strong match with learned category rules.

To more explicitly test whether our multivoxel results on the ambiguous trials support the representational assumptions of distinct mechanisms that contribute to common versus rare responses, we computed Bayes factors to evaluate the strength of evidence for and against the null hypothesis in both cases. According to the predictions of dual-process accounts, no differences in mean pattern similarity to rare versus common stimulus dimensions would be expected for common responses, as participants are expected to retrieve weak or competing representations of the category exemplars in these cases and thus respond in a way consistent with the category base rates. Alternatively, enhanced pattern similarity to common stimulus dimensions would be expected in the case of rare responses, in line with dissGCM's explanation of the IBRE. Bayes factors for each of these hypotheses were tested using the BIC approximation method, computing $\exp(\Delta BIC/2)$ between the null and alternative models (*Wagenmakers, 2007*). The resulting Bayes factors suggested positive evidence in favor of the null hypothesis for pattern similarity on common response trials ($BF_{01} = 7.54$), and conversely, positive evidence in favor of the alternative hypothesis that common features would be more strongly represented on rare response trials ($BF_{10} = 4.36$). Accordingly, beyond revealing that a dissimilarity-based process contributes to rare responding in an IBRE task, our multivoxel results

point to the existence of a distinct process for common responses that is, on average, less dependent on the activation of common or rare category exemplars. Our behavioral findings provide additional evidence for such a dissociation, as reaction times were found to be significantly slower for rare relative to common responses.

## Learning phase

Beyond our primary questions about test phase activation, multivoxel analysis of the learning phase can provide additional information about how participants processed stimuli in the present task. Generally, both similarity-based models and dissimilarity-based models such as the dissGCM predict that features which are most informative about the correct category will contribute more to categorization decisions during learning. With respect to multivoxel predictions, this means that activation patterns elicited during learning should contain more information about the predictive features (objects or scenes) than non-predictive features (faces), and both of these types of information should be activated more strongly than non-present features. *Figure 4* depicts mean pattern similarities for predictive, non-predictive, and non-present visual stimulus categories during the learning phase for both common and rare disease trials. As anticipated, a linear mixed effects model collapsed across trial type revealed that pattern similarity to the visual category was the strongest for perfectly predictive features (M = .065), followed by the non-predictive but present features (M = −0.050) and the non-present features (M = −0.145), ($F$ (2, 42)=54.8, p<0.001). This finding, whereby activation patterns elicited for stimuli during learning are most similar to predictive features, is consistent with recent studies using MVPA to measure dimensional selective attention in categorization and reinforcement learning (*Mack et al., 2013*, *Mack et al., 2016*; *Leong et al., 2017*;

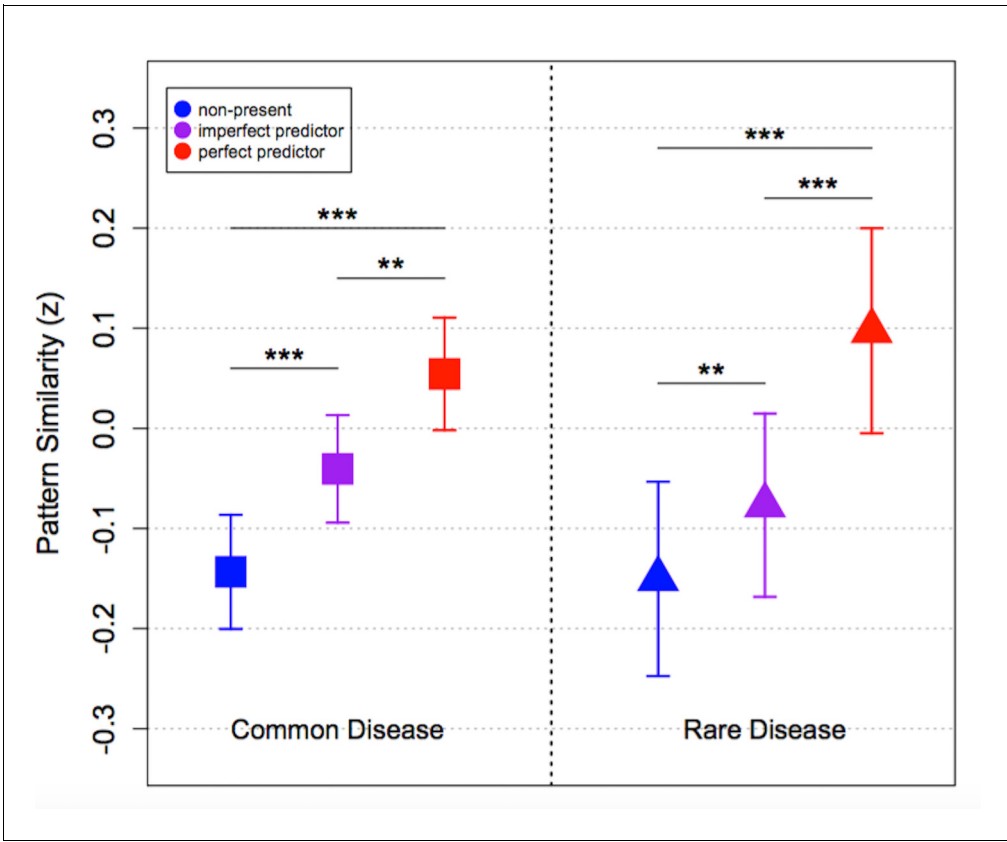

**Figure 4.** Multivoxel pattern similarity to each feature type during the learning phase (mean ± SEM). The left panel is for trials predictive of a common disease, and the right for trials predictive of a rare disease. Red points represent perfect predictors, purple points represent imperfect predictors (faces), and blue points represent non-present features. **p < 0.01, ***p < 0.001.
DOI: https://doi.org/10.7554/eLife.36395.008

*O'Bryan et al., 2018*). For common trials, pairwise comparisons revealed significant differences between pattern similarity to perfect and imperfect predictors ($t$ (21)=3.38, p=0.003), perfect predictors and non-present features ($t$ (21)=5.71, p<0.001), and between imperfect predictors and non-present features ($t$ (21)=4.27, p<0.001). Likewise, for rare trials we found significant differences between pattern similarity to perfect and imperfect predictors ($t$ (21)=5.69, p<0.001), perfect predictors and non-present features ($t$ (21)=9.11, p<0.001), and between imperfect predictors and non-present features in the expected directions ($t$ (21)=3.11, p=0.005) (see *Figure 4*).

Although the greater contribution of predictive features to multivoxel activation patterns during learning is a straightforward prediction that is consistent with any model, a further question is how such activation patterns during learning contribute to later test performance. As with the test phase, dissimilarity-based theories make the somewhat counterintuitive prediction that it is specifically what people learn about the common category that is driving later choices of the rare category. Mechanistically, dissimilarity-based generalization is thought to involve a comparison process whereby memory-based representations of learned category exemplars are retrieved and then contrasted with the stimulus currently under consideration. Because of the base rate manipulation in an IBRE task, people are expected to retrieve information about the common category more readily when faced with ambiguous transfer stimuli, thus making them more likely to choose the rare category in such cases via elimination (*Juslin et al., 2001*; *Winman et al., 2005*).

Because the dissGCM is designed to explain group-level generalization patterns rather than individual differences per se, the model incorporates base rate information using exemplar-specific memory strength parameters ($t_j$), which we fix at the true 3:1 category base rates across subjects. However, a realistic expectation is that subjects will differ in the extent to which they learn (and eventually recall) the predictive values associated with common and rare exemplars. Here, our pattern similarity measure provides an opportunity to investigate whether individuals who more strongly represent information associated with the common category over the course of learning neglect the category base rates more frequently at test, as predicted by dissimilarity-based theories of IBRE. This prediction is in direct contrast to the dominant similarity-based explanation of IBRE, which posits that it is specifically a stronger learned association between rare perfect predictors and their category that drives later rare category selections for the ambiguous test probes (*Medin and Edelson, 1988*; *Kruschke, 1996*, *Kruschke, 2001*). Accordingly, we tested these hypotheses by computing Pearson correlations between mean pattern similarity to each stimulus dimension during the learning phase and subjects' choice behavior on the critical ambiguous trials.

*Figure 5* depicts the associations between BOLD pattern similarity to common and rare stimulus dimensions during learning and base rate neglect. Consistent with dissimilarity-based accounts, we found that greater activation of multivoxel patterns associated with common perfect predictors during learning was correlated with a higher proportion of rare choices on the ambiguous test trials ($r = 0.590$, $t$ (20)=3.27, p=0.004). Alternatively, we found no significant relationship between activation of multivoxel patterns during learning and choice proportions on IBRE trials for rare perfect predictors ($r = 0.114$, $t$ (20)=0.512, p=0.614), common faces ($r = -0.163$, $t$ (20)=-0.737, p=0.470), or rare faces ($r = -0.039$, $t$ (20)=-0.173, p=0.865). A linear mixed effects model including factors for each item and participants' rare choice proportions was used to test for significant differences in slope among the above associations. The model revealed an interaction between pattern similarity to different learning phase items and base rate neglect ($F$ (3, 60)=2.97, p=0.039). Specifically, the relationship between pattern similarity to common perfect predictors and base rate neglect was stronger, in the positive direction, than those for common faces ($t$ (60)=2.72, p=0.009), and rare faces ($t$ (60)=2.40, p=0.020). Likewise, the difference in positive slope between pattern similarity and base rate neglect for common versus rare perfect predictors was marginally significant ($t$ (60)=1.88, p=0.065).

## Model-based univariate results

By revealing a link between activation of common feature patterns and the IBRE, our multivoxel results suggest that dissimilarity-based evidence against unchosen categories contributes to choice behavior in the present task. However, it remains an open question whether such dissimilarity processes involve distinct neural or cognitive mechanisms beyond those thought to underlie basic similarity processes. Importantly, similarity-based theories propose that a single, non-inferential cognitive process is responsible for generalization patterns across trials in the IBRE task (*Medin and*

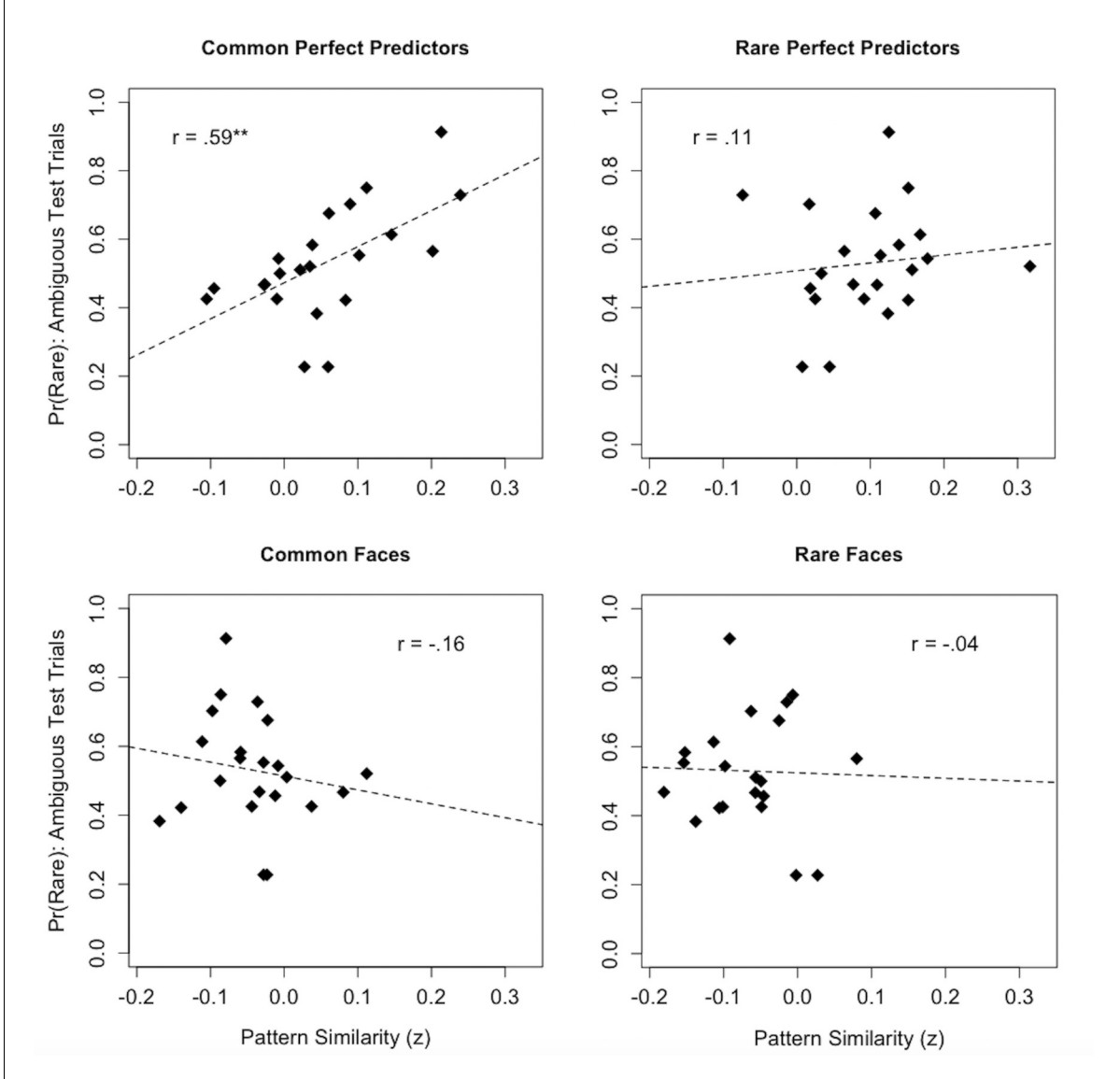

**Figure 5.** Associations between multivoxel pattern similarity to stimulus dimensions during the learning phase and individual differences in base rate neglect. For each graph, the y-axis depicts the proportion of rare responses made by each subject on ambiguous test trials, while the x-axis depicts subjects' mean BOLD pattern similarity to a respective stimulus dimension over the course of learning. **p < 0.01.
DOI: https://doi.org/10.7554/eLife.36395.009

*Edelson, 1988*; *Kruschke, 1996*, *Kruschke, 2001*), and thus it is anticipated that a network of brain regions associated with similarity-based generalization underlies choice across task contexts in the present study. Although the dissGCM is agnostic as to whether using dissimilarity as evidence is more cognitively demanding than relying on similarity alone, previous theories of IBRE positing dissimilarity processes propose that the use of contrastive evidence is inherently inferential (*Juslin et al., 2001*; *Winman et al., 2005*). Accordingly, the latter account would predict a unique neural topography associated with dissimilarity-based evidence, including regions known to be involved in higher-level, symbolic reasoning.

To test whether similarity- and dissimilarity-based evidence rely on different brain regions, we modeled univariate voxel-wise activation using trial-by-trial estimates of similarity- and dissimilarity-based evidence derived from the dissGCM. Specifically, the overall evidence *v* for the winning category on each test trial was decomposed into two separate regressors: one for summed similarity to the winning category, and the other for summed dissimilarity to the non-winning categories. The

regions associated with dissimilarity-based evidence in this analysis are thus distinct from those negatively associated with similarity-based evidence because they are derived from evidence against the alternative, non-winning category.

Our analysis showed that greater similarity-based contributions to the winning category were associated with activation in the MTL (left hippocampus) vmPFC, and primary motor cortex (*Figure 6A*, depicted in red; *Table 2*). These results are consistent with findings from other model-based fMRI studies suggesting that the MTL is involved in similarity-based retrieval (*Davis et al., 2012a*, *Davis et al., 2012b*). Likewise, the engagement of vmPFC corroborates recent studies suggesting that this region tracks higher relative evidence for categorization decisions (*Davis et al., 2017*; *O'Bryan et al., 2018*). The positive relationship between vmPFC and similarity processes may also be reflective of attention to strong predictors (*Sharpe and Killcross, 2015*; *Nasser et al., 2017*)

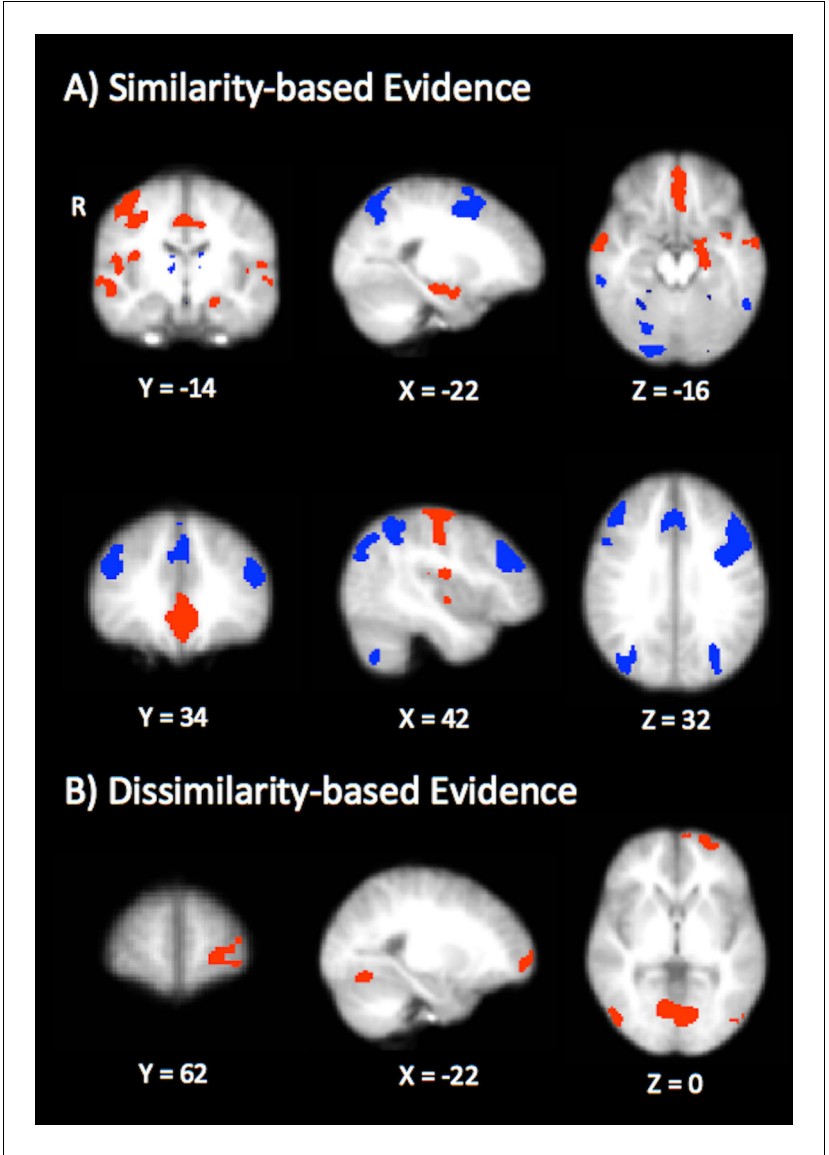

**Figure 6.** Results from the model-based univariate analysis. (**A**) Depicts activation that tracks similarity-based contributions to choice (summed similarity to the winning category). Red depicts activation positively correlated with similarity-based contributions and blue depicts negatively correlated activation. (**B**) Depicts brain regions that are positively correlated with dissimilarity-based contributions to choice (summed dissimilarity to the non-winning category).
DOI: https://doi.org/10.7554/eLife.36395.010

**Table 2.** Activated clusters and peaks for the fMRI results in *Figure 6*.

| Contrast | Regions | Peak *t*-value | Peak MNI coordinates (x,y,z) | Number of voxels | Cluster *P* |
|---|---|---|---|---|---|
| Similarity > 0 | | | | | |
| | Rostral and ventral medial prefrontal cortex | 7.76 | 0, 54, −2 | 2268 | p<0.001 |
| | Middle temporal gyrus | 5.74 | 64, −6, −14 | 1050 | p=0.016 |
| | Precentral gyrus | 7.65 | 42, −16, 62 | 1031 | p<0.001 |
| | Precentral gyrus | 4.81 | 0, −30, 58 | 463 | p=0.008 |
| | Middle temporal gyrus | 4.98 | −62, −2, −16 | 322 | p=0.016 |
| | Parietal operculum cortex | 5.74 | −34, −30, 18 | 282 | p=0.027 |
| | Hippocampus | 5.25 | −22, −18, −16 | 271 | p=0.019 |
| | Lateral occipital cortex (inferior) | 4.68 | 50, −72, 12 | 157 | p=0.039 |
| Similarity < 0 | | | | | |
| | Superior parietal and lateral occipital cortex (superior) | 8.43 | −44, −46, 58 | 5123 | p<0.001 |
| | Middle frontal gyrus | 7.01 | −52, 12, 36 | 2491 | p<0.001 |
| | Dorsal medial PFC | 9.00 | −2, 18, 46 | 917 | p<0.001 |
| | Cerebellum | 6.11 | 28, −64, −28 | 769 | p=0.002 |
| | Middle frontal gyrus | 6.23 | 32, 2, 62 | 730 | p=0.002 |
| | Middle frontal gyrus | 5.89 | 44, 36, 30 | 391 | p=0.009 |
| | Inferior temporal gyrus | 6.09 | −54, −52, −12 | 375 | p=0.009 |
| | Inferior temporal gyrus | 6.53 | 60, −52, −10 | 271 | p=0.016 |
| | Cerebellum | 6.08 | −28, −60, −32 | 186 | p=0.031 |
| | Thalamus | 4.88 | −10, −18, 10 | 175 | p=0.038 |
| Dissimilarity > 0 | | | | | |
| | Occipital cortex | 7.55 | 12, −78, 12 | 1372 | p<0.001 |
| | Fusiform and lateral occipital cortex (inferior) | 6.82 | 42, −60, −18 | 865 | p=0.002 |
| | Fusiform and lateral occipital cortex (inferior) | 5.97 | −36, −52, −18 | 575 | p=0.003 |
| | Middle frontal gyrus | 5.00 | 54, 16, 34 | 255 | p=0.020 |
| | Frontal pole (rostrolateral PFC) | 5.93 | −42, 52, −6 | 130 | p=0.047 |

DOI: https://doi.org/10.7554/eLife.36395.011

or the application of familiar category rules (*Boettiger and D'Esposito, 2005*; *Liu et al., 2015*), both of which are consistent with similarity-based accounts of IBRE that attribute choice to a well-established association between perfect predictors and their outcomes that is driven by attention (e.g. *Kruschke, 1996*). We also found that greater contributions of similarity-based evidence were positively associated with activation in the primary motor cortex. While more anterior motor-planning regions such as pre-SMA and SMA tend to be associated with rule acquisition processes (e.g. *Boettiger and D'Esposito, 2005*), primary motor cortex has been found to track increasing levels of response automaticity in categorization tasks (*Waldschmidt and Ashby, 2011*).

The dlPFC, dorsomedial PFC, and posterior parietal cortex were found to be negatively correlated with similarity-based evidence for the chosen category (*Figure 6A*, depicted in blue; *Table 2*). This fronto-parietal network is generally associated with rule-based category learning (*Filoteo et al., 2005*; *Seger and Cincotta, 2006*; *Soto et al., 2013*), and is thought to play a critical role in representing the uncertainty associated with categorization decisions (*DeGutis and D'Esposito, 2007*; *Seger et al., 2015*; *Davis et al., 2017*). Thus, our results are consistent with these findings, and moreover, suggest that dlPFC and functionally related fronto-parietal regions may be engaged in cases where probes fail to elicit a strong similarity-based match with stored category exemplars.

Contrastingly, dissimilarity-based evidence was positively correlated with activation in the left rlPFC (*Figure 6B*; *Table 2*), consistent with our hypothesis that this type of evidence might encourage more symbolic processes believed to underlie the rlPFC's contribution to category learning (*Davis et al., 2017*; *Paniukov and Davis, 2018*). No clusters were significantly negatively associated with dissimilarity-based evidence. Comparing the statistical maps in *Figure 6A and B*, it is apparent that greater relative contributions of dissimilarity-based evidence do not necessitate smaller contributions of similarity-based evidence in the dissGCM: if these regressors were anticorrelated, one would expect the regions associated with dissimilarity processes to resemble the fronto-parietal network we found to be negatively associated with similarity. Instead, our results show that using contrastive evidence uniquely engages the rlPFC, in line with dual-process theories that suggest dissimilarity-based processing may distinctly depend on higher-level, abstract reasoning. Despite obvious differences between the activation patterns elicited for each contrast, it is notable that all three maps (positive similarity, negative similarity, and positive dissimilarity) revealed significant activation in portions of ventral occipitotemporal cortex. These regions (lateral occipital cortex, inferior temporal gyrus, and fusiform gyrus) have well-established roles in representing visual object categories, including those used as stimulus dimensions in the present study (*Grill-Spector and Weiner, 2014*). Accordingly, it is possible that the engagement of these regions in our study reflects feature-based attention, exemplar retrieval, or a combination of both processes that occurs regardless of the respective contributions that similarity- and dissimilarity-based evidence make to a decision.

## Discussion

The present study employed model-based fMRI to test how similarity and dissimilarity contribute to the inverse base rate effect (IBRE) and how these types of evidence relate to neural mechanisms that support category learning. The dominant theory behind the IBRE suggests that it arises from attentional processes that make ambiguous items containing features of rare and common categories seem more similar to members of the rare category. Here we find support for the hypothesis that dissimilarity-based evidence also contributes to the IBRE: people may categorize the ambiguous stimuli as members of the rare category not only because of their similarity to the rare category, but also because of their dissimilarity to members of the common category.

The dissGCM, an extension of the GCM that allows for the use of dissimilarity-based evidence in categorization behavior, predicted two novel observations in the neuroimaging data. First, as predicted by the dissGCM's relative contribution of similarity- and dissimilarity-based evidence during the ambiguous trials, multivoxel analysis suggested stronger activation of patterns associated with features of the common category when participants classified ambiguous stimuli as rare. Second, model-based univariate analysis revealed that measures of similarity- and dissimilarity-based evidence had unique neural topographies. Similarity-based evidence for the winning category was positively correlated with regions of the hippocampus, vmPFC, and primary motor cortex. In contrast, dlPFC, dorsomedial PFC, and posterior parietal cortex were negatively correlated with similarity-based contributions. Dissimilarity-based evidence against non-winning categories was positively correlated with the left rlPFC.

The present results raise several important questions about the cognitive and neural mechanisms underlying people's use of base rate information. Previous theories arguing for dissimilarity-like processes as explanations of IBRE have argued that they arise from mechanisms rooted in higher-level propositional logic that fundamentally differ from the similarity-based mechanisms posited by dominant theories (*Juslin et al., 2001*). As illustrated by the dissGCM, such dissimilarity-based processes can be viewed as simple extensions of similarity-based processing and need not depend on the existence of a functionally separate categorization system. At the same time, our neuroimaging results suggest that dissimilarity, but not similarity-based evidence may arise from processing in rlPFC regions that are known to be involved with higher-level reasoning and problem solving (*Christoff et al., 2001*; *Bunge et al., 2005*, *Bunge et al., 2009*). One possibility for reconciling these theories is that the dissimilarity-based evidence involves more abstract or symbolic feature processing than pure similarity processes, and this additional processing taps rlPFC regions. This is consistent with our recent model-based fMRI results, which demonstrate that rlPFC tracks measures of relational encoding in category learning, but otherwise this type of category learning may rely on the same basic similarity-based mechanisms as simpler feature-based learning (*Davis et al., 2017*).

By establishing that the rlPFC is engaged when participants incorporate dissimilarity-based evidence into categorization decisions, our research adds to a growing literature aiming to pinpoint a domain-general computational role for this region. A common thread among tasks shown to engage the rlPFC is that they tend to involve combining across disparate representations to form the basis for a decision – whether those representations comprise confidence estimates and subjective value (*De Martino et al., 2013*), visual features and their relations (*Bunge, 2004*; *Bunge et al., 2009*; *Davis et al., 2017*), or expected rewards and their relative uncertainties (*Boorman et al., 2009*; *Badre et al., 2012*). Likewise, in the case of the current study, the evidence that an ambiguous stimulus is similar to a given category must be combined with the evidence that the stimulus is dissimilar to the other possible categories. Although the dissGCM instantiates dissimilarity as a simple transformation of similarity, the involvement of rlPFC when participants place more reliance on dissimilarity-based evidence may be attributable to increasing demands for integrating evidence across several abstract representations. A decision made on pure similarity-based evidence would require no such integration. This hypothesis accords with recent findings implicating the rlPFC in evaluative processes for categorization tasks that require candidate rules to be weighed over the course of several trials, relative to matching tasks where a rule can be known with certainty following a single correct trial (*Paniukov and Davis, 2018*).

One question that has arisen repeatedly in the literature on the IBRE is whether it reflects an inherent irrationality in decision making. When viewed through the lens of basic similarity-based attentional processes (e.g. *Medin and Edelson, 1988*; *Kruschke, 1996*, *Kruschke, 2001*), the IBRE appears to arise from very simple learning mechanisms that are not particularly tied to higher-level rationality, and rare choices seem to indicate a lack of knowledge of the base rates. Indeed, in a separate model fit, we attempted to fit the standard similarity-based GCM to the key pattern on the ambiguous trials. However, the standard GCM was only able to predict a greater proportion of rare choices if accurate knowledge of the exemplar base rates was eliminated (all values of $t_j = 1$ or fit as free parameters). In contrast, accurate knowledge of the category base rates directly contributes to the greater dissimilarity-based evidence against the common category. Thus from the dissGCM perspective, participants are perfectly knowledgeable about the base rates in the present task, but they use this knowledge in a way not anticipated by pure similarity-based models. However, whether or not this use of dissimilarity-based evidence constitutes irrationality is a deeper question that cannot be answered based purely on the present results.

How or whether the use of dissimilarity is encouraged by the standard IBRE design, compared with other types of categorization problems, is an open question. The dissGCM was originally developed to explain sequential effects in categorization (*Stewart and Morin, 2007*), and its success in this domain suggests that dissimilarity processes, such as those revealed here, may be present in many categorization tasks that are more familiar in the neuroimaging literature. However, how much a task encourages dissimilarity-based processing may vary considerably and depend on a number of factors. For example, purely attentional accounts posit that strong initial learning of the common category leads people to learn the rare category by the features that distinguish it from the common (*Markman, 1989*; *Kruschke, 1996*). Learning-order effects do appear to play a role in the IBRE: previous studies have shown that using blocked or unequally distributed category frequencies during training leads participants to favor later-learned categories on ambiguous test probes, even when overall base rates are held constant (*Medin and Bettger, 1991*; *Kruschke, 1996*).

Early studies on the role of order and blocking on IBRE are similar to current work on blocked versus interleaved learning in the broader categorization literature (*Birnbaum et al., 2013*; *Carvalho and Goldstone, 2014*, *Carvalho and Goldstone, 2015*; *Goldwater et al., 2018*). In blocked learning, categories are learned by viewing a number of items from the same category before switching to other categories. For example, in the present case, if we had used blocked learning, participants may see an entire block of Disease 1 examples, and then blocks of Disease 2, 3, and 4, but the examples of the Diseases would not be intermixed. In interleaved learning, the standard for category learning, items from all categories are presented in a random order such that the examples of the different categories are intermixed. Blocked learning tends to lead participants to focus more on stimulus features that are shared with members of the same category, whereas interleaved learning tends to lead participants to focus more on features that differentiate categories. Interestingly, while not an interleaved versus blocked manipulation per se, frequency manipulations such as those used in the present study have an effect of creating more blocking within common

categories – common categories are more likely to follow examples of the same category, and inter-leaving within rare categories. Although discovered long after the initial IBRE studies, blocked versus interleaved learning theories may offer a concurring explanation of the IBRE that does not depend on differences in the rate at which common and rare categories are learned. However, formal computational models of blocked versus interleaved learning have thus far focused on how these scenarios produce differences in selective attention to stimulus features that are characteristic (blocked) or diagnostic (interleaved) of a category, and are pure similarity-based models such as the original GCM (*Carvalho and Goldstone, 2017*). Contrastingly, our MVPA and univariate fMRI results show that pure similarity-based processing cannot fully explain the IBRE, and thus strongly suggest that dissimilarity processes contribute to the IBRE.

To investigate how learning manipulations, such as blocked versus interleaved, or individual differences in learning influence the mechanisms we propose here, it will be critical for future research to build full learning models of the dissGCM. The dissGCM, like the standard GCM, is a model of asymptotic categorization performance and generalization, and thus is not well-equipped to account for learning dynamics or individual differences. For these reasons, our individual difference analysis focused on using MVPA estimates from individual stimuli rather than formal model-based analysis. Nonetheless, these analyses reveal important individual differences that are consistent with dissimilarity-based theories more broadly. Dissimilarity-based theories posit that one of the reasons IBRE arises is because the common category becomes more thoroughly established in memory during learning, which leads participants to retrieve this information more readily at test. From this perspective, participants who learn the common category more strongly should consequently exhibit base rate neglect more frequently. Consistent with these predictions of dissimilarity-based theories, pattern similarity analyses revealed that participants who more strongly activated information associated with common categories during learning engaged in base rate neglect more often at test. While these results suggest that individual differences in learning contribute to IBRE, they nonetheless point to a critical need to develop a full learning-based version of the dissGCM that can be applied at the individual level to capture these differences. For example, one possibility is that participants' weighting of similarity and dissimilarity (the $s$ parameter) changes over learning based on the participants' learning rates and factors related to blocking versus interleaved presentation (*Carvalho and Goldstone, 2017*). However, such a model would require extensive additional data to validate, and thus is beyond the scope of the present study.

The IBRE exemplifies a case in cognitive neuroscience where independent models that predict essentially the same behavioral patterns make very different assumptions about the cognitive processes, and accordingly, brain states, involved in producing the behavior. Our findings from the test phase represent a critical step forward in an emerging area of research using multivariate fMRI to reveal that qualitatively distinct brain states may reflect the use of multiple response strategies in the face of identical stimuli (e.g. *Mack et al., 2013*). Consistent with past research using MVPA to decode learned selective attention (*Mack et al., 2013*, *Mack et al., 2016*; *Leong et al., 2017*; *O'Bryan et al., 2018*), multivoxel patterns associated with predictive features were more strongly activated than imperfectly predictive features during the learning phase. Using the same approach to decode which information participants were focusing on during ambiguous test trials, we found stronger activation of patterns associated with common compared with rare stimulus features, but importantly, this pattern only emerged in cases where participants chose the rare category. Moreover, rare category selections were accompanied by slower RTs relative to common selections. These results are consistent with a higher-level, dissimilarity-based process where activating information associated with common exemplars provides contrastive evidence against the well-established common category. Alternatively, it is possible that participants are more likely to respond according to the base rates when the ambiguous stimuli elicit a strong similarity-based match: given our RT results along with the correlation between similarity-based evidence and motor cortex engagement, in these cases subjects may revert to habitual response patterns from the learning phase and simply choose the more well-established (common) category. However, understanding the precise cognitive mechanisms that contribute to these response-dependent activation patterns remains a direction for future research.

Interestingly, while our findings argue against the prediction from similarity-based models that the IBRE arises because rare features become more similar to their associated category, the observed attention weight parameters $w_k$ from the model fits are consistent with a key part of

similarity theory – that there is greater selective attention allocated to the rare feature dimension. Indeed, the rare feature dimensions outweighed the common features for both sets of categories in our data. However, these larger attention weights did not seem to drive greater pattern similarity to the rare feature dimension in our multivoxel results. We predict that our multivoxel results are not driven directly by simple feature-based attention, but instead indicate some combination of attention and memory-based retrieval of the category exemplars. Pattern similarity measures in ventral temporal cortex have been shown to effectively index both dimensional selective attention (*Leong et al., 2017*; *O'Bryan et al., 2018*) and the retrieval of non-present, associated stimuli (e.g. *Zeithamova et al., 2012*; for review, see *Rissman and Wagner, 2012*). Rather than adjudicating between whether the multivoxel patterns in the current study are more likely to indicate attention or memory, a possibility that accords with both potential explanations is that these pattern similarity indices reflect information that is actively represented in working memory, either by way of visual cueing or reinstated long-term memories (*Lewis-Peacock and Postle, 2008*). In cases in which multiple or competing stimulus representations are present in WM, as may be expected for the ambiguous IBRE trials, multivoxel patterns should be most similar to whichever representation is consciously attended (*Lewis-Peacock et al., 2012*). However, given the design of the current study we are unable to rule out the possibility that implicitly activated or post-decisional feature representations contribute to our pattern similarity results. Future studies may wish to combine multivoxel pattern analysis with eye-tracking (e.g. *Leong et al., 2017*) to better understand the unique contributions that attention and memory make to the present results.

In conclusion, using model-based fMRI analysis, we found evidence that extreme cases of base rate neglect such as the IBRE may arise from a combination of similarity- and dissimilarity-based processes. Accordingly, measures of neural activation suggest that people may be more strongly relying on evidence about how dissimilar an item is to common categories when faced with ambiguous stimuli. Furthermore, dissimilarity processes have a unique cortical topography that includes the rostrolateral PFC, a region believed to be involved with more symbolic feature processing.

## Materials and methods

Twenty-four healthy right-handed volunteers (age range 18–58; 13 women) participated in the study for \$35. All protocols were approved by the Texas Tech University IRB. Two participants were excluded, one for falling asleep and the other for registration failures in the first five scanning runs.

### Behavioral protocol

The study consisted of three phases: localizer, learning, and test. The localizer phase consisted of two scanning runs (run length = 5 min 10 s) in which participants classified images based on whether they contained a face, an object, or a scene. Each image was presented for 2.5 s during which participants were asked to respond 'Scene (1), Face (2), or Object (3)?' Each trial was separated with random fixation drawn from a truncated exponential distribution with mean = 3 s. Over the duration of the localizer phase, subjects categorized 38 examples of each stimulus type. The face, object, and scene images used were black-and-white squares presented on a white background with black text. The stimuli used during the localizer runs were presented in a random order, and did not include any of the images used for the experimental task.

In the learning phase, participants learned a classic IBRE category structure (*Medin and Edelson, 1988*; see *Figure 1*). The features used for the stimuli included examples of faces, objects, and scenes not shown in the localizer phase. Participants were given an epidemiological cover story asking them to predict whether hypothetical patients would contract a disease based on the people they have been in contact with (faces), the objects they have used (objects), and the places they have been (scenes). On each trial of the learning phase, participants would see a stimulus for 3 s and were asked to answer 'Disease 1, 2, 3, or 4?' This was followed by random fixation, feedback (1.75 s) in which they were told whether they were right or wrong and the correct answer, and additional fixation. The same distribution was used to generate fixations as in the localizer phase. Faces were always assigned to the imperfectly predictive feature dimensions, whereas objects and scenes were perfectly predictive and associated with only one disease (*Figure 1*). To ensure that no visual stimulus category differed in overall frequency, one common disease was always associated with objects and the other scenes, and likewise for rare diseases. Participants were randomly assigned to one of

two conditions to balance which images were presented together during learning and test, and disease labels were randomized across participants. Within-pair stimulus position (left or right) was randomized on each trial, and the presentation order of feature pairs was randomized within each block for every participant. The learning phase was spread over three scanning runs (run length = 5 min 10 s), and four blocks of the stimulus set were presented per run, resulting in a total of 12 blocks and 96 trials for the learning phase. The progression of a learning trial is depicted in the bottom panel of *Figure 1*.

During the test phase, participants completed trials with both new and old exemplars and classified them as 'Disease 1, 2, 3, or 4?', but no longer received feedback. New items included all possible single and two-feature combinations of the perfectly predictive features (see *Table 1*, Results). Trials were 3 s and separated by random fixation as described above. Like the learning phase, the test phase occurred over three consecutive scanning runs (run length = 5 min 10 s). Each item in the stimulus set was encountered twice per run, with the exception of the ambiguous perfect predictor pairs which were repeated four times per run. This resulted in 24 instances of ambiguous scene-object pairs, and 48 instances of the ambiguous trials overall for each participant. Presentation order of the test items was randomized for each of the three runs, with participants rating two test sets per run, resulting in a total of 156 test trials.

## Model

The dissimilarity generalized context model (dissGCM; *Stewart and Morin, 2007*) is an extension of the generalized context model (*Nosofsky, 1986*) that accounts for choice using a combination of similarity- and dissimilarity-based evidence. Like the original GCM, stimuli are represented as points in a multidimensional feature space. The model computes distances in this space between probe stimuli $S_i$ and stored exemplars $S_j$ along each dimension $k$:

$$d_{ij} = \left( \sum_{k=1}^{K} w_k |S_{ik} - S_{jk}|^r \right)^{1/r}, \qquad (1)$$

where $r$ defines the metric of the space, here assumed to be one (city-block). The $w_k$ indicates dimensional attention weights, which have the function of stretching the distance along strongly attended dimensions, and are constrained to sum to one.

Distances are converted to similarities via an exponential transform:

$$sim_{ij} = e^{-cd_{ij}}, \qquad (2)$$

where $c$ is a specificity parameter that controls the rate at which similarity decays as a function of distance.

The first contribution of evidence for a given category comes from the summed similarity between a probe and all stored exemplars for that category, consistent with the original GCM. DissGCM then combines this similarity-based contribution with the summed *dissimilarity* between a probe and the exemplars from all other categories. The overall evidence, $v$, for a category $C_A$, given stimulus $S_i$ is:

$$v_{iA} = s \sum_{s_j \in C_A} t_j sim_{ij} + (1-s) \sum_{s_j \in \neg C_A} t_j (1 - sim_{ij}), \qquad (3)$$

where $s$ is a free parameter that determines how much the model weights similarity versus dissimilarity. The parameter $t_j$ reflects exemplar-specific memory strength, which we fix at each exemplar's true base rate during learning (1 for rare category exemplars, 3 for common category exemplars). Here, we also make the assumption that exemplars only contribute evidence (similarity or dissimilarity) if they have at least one positive feature match with a probe stimulus.

The model makes a prediction for how likely an item is to be classified as a member of a given category $C_A$ by:

$$pr(resp = C_A | S_i) = \frac{v_{iA} + b}{\sum v_{iC} + 4b}, \qquad (4)$$

where $b$ is a free parameter that reflects the baseline level of similarity for a category that has

0 positive feature matches. More generally, this parameter ensures that no predicted probabilities are 0 or 1, which interferes with the maximum likelihood-based model fits.

The model was fit to the group response frequencies for each option by minimizing the −2 * Log Likelihood using a differential evolution function optimizer. The overall fit was 4,314.588. The best fitting parameters for each of the dimension weights were $w_1$ (face 1)=0.277, $w_2$ (common scene) =0.665, $w_3$ (rare object)=0.887, $w_4$ (face 2)=0.170, $w_5$ (common object)=0.712, and $w_6$ (rare scene) =0.879); $c$ = 9.05; $s$ = 0.946; $b$ = 0.023.

## Image acquisition

Imaging data were acquired on a 3.0 T Siemens Skyra MRI scanner at the Texas Tech Neuroimaging Institute. Structural images were acquired in the sagittal plane using MPRAGE whole-brain anatomical scans (TR = 1.9 s; TE = 2.44 ms; $\theta$ = 9°; FOV = 250 × 250 mm; matrix = 256 × 256 mm; slice thickness = 1.0 mm, slices = 192). Functional images were acquired using a single-shot T2*-weighted gradient echo EPI sequence (TR = 2.5 s; TE = 25 ms; $\theta$ = 75°; FOV = 192 × 192 mm; matrix = 64 × 64; slice thickness = 3 mm).

## fMRI analysis and preprocessing

Functional data were preprocessed and analyzed using FSL (www.fmrib.ox.ac.uk/fsl). Anatomical images were preprocessed using Freesurfer (autorecon1). Functional images were skull stripped, motion corrected, prewhitened, and high-pass filtered (cutoff: 60 s). For the model-based univariate analysis, functional images were spatially smoothed using a 6 mm FWHM Gaussian kernel. No smoothing was performed on functional data used for the multivoxel analysis. First-level statistical maps were registered to the Montreal Neurological Institute (MNI)−152 template using 6-DOF boundary-based registration to align the functional image to the Freesurfer-processed high-resolution anatomical image, and 12-DOF affine registration to the MNI-152 brain.

## Model-based univariate analysis

The model-based univariate analysis employed a standard three-level mixed effects model carried out in FSL's FEAT program. The first-level model included an EV for stimulus presentation and two model-based parametric modulators: similarity- and dissimilarity-based evidence, computed from the dissGCM. Specifically, these regressors were obtained on a trial-by trial basis using *equation 3* (see Model section), where the evidence contribution of summed similarity to the winning category ($C_A$; most probable category according to the model) is calculated as:

$$s \sum_{S_j \in C_A} t_j sim_{ij}, \tag{5}$$

and the evidence contribution of summed dissimilarity to non-winning categories with a positive feature match is calculated as:

$$(1-s) \sum_{S_j \in \neg C_A} t_j \left(1 - sim_{ij}\right), \tag{6}$$

Both parametric modulators were centered and scaled (z-scored) within run. Additional explanatory variables (EVs) of no interest included motion parameters, their temporal derivatives, EVs to censor volumes exceeding a framewise displacement of 0.9 mm (*Siegel et al., 2014*), and an EV to account for trials in which participants failed to make a behavioral response. Final statistical maps were corrected for multiple comparisons using a non-parametric cluster-mass-based correction with a cluster-forming threshold of $t$ (21)=3.52 (p<0.001, one-tailed).

## Multivoxel pattern analysis

RSA was conducted using the PyMVPA toolbox (*Hanke et al., 2009*) and custom Python routines. To obtain trial-by-trial estimates of the hemodynamic response, we computed a β-map (*Rissman et al., 2004*) for each stimulus onset using an LS-A procedure (*Mumford et al., 2012*), simultaneously modeling the trials of interest as separate regressors in a GLM. These estimates were anatomically restricted to three ventral temporal ROIs that were maximally responsive to scene, object, and face information in the localizer data. Specifically, pattern estimates were spatially

localized in visual stimulus category-specific ROIs by creating 6 mm spheres around subjects' peak activation within anatomically defined regions in the Harvard-Oxford Atlas associated with category selectivity (objects: left inferior posterior temporal gyrus; scenes: bilateral parahippocampal gyrus; faces: right temporal occipital fusiform gyrus; *Ishai et al., 1999*; *Lewis-Peacock and Postle, 2008*; *Lewis-Peacock et al., 2012*; *Grill-Spector and Weiner, 2014*). The last trial of each run was automatically discarded from the multivoxel analysis to ensure stable estimation of the activation patterns for all trials. Additional explanatory variables (EVs) of no interest included motion parameters, their temporal derivatives, and EVs to censor volumes exceeding a framewise displacement of 0.9 mm.

For the primary pattern similarity analyses, we measured how much participants were activating scene, object, and face information on individual test phase trials by calculating mean correlation distance (1 − Pearson's $r$) between activation patterns on each test trial and those elicited for each visual category during the localizer phase. For interpretative ease, the distances were converted to similarities using exp(- distance), and then standardized ($z$-scored) within participants. Source data and scripts used to create all figures and tables (e.g. R code, PyMVPA scripts, statistical maps for the model-based fMRI analysis) are freely available online at https://osf.io/atbz7/.

## Acknowledgements

This work was supported by start-up funds to TD from Texas Tech University. The authors declare no conflicts of interest.

## Additional information

### Funding

| Funder | Author |
| --- | --- |
| Texas Tech University | Tyler Davis |

The funders had no role in study design, data collection and interpretation, or the decision to submit the work for publication.

### Author contributions

Sean R O'Bryan, Conceptualization, Data curation, Software, Formal analysis, Validation, Investigation, Visualization, Methodology, Writing—original draft, Project administration, Writing—review and editing; Darrell A Worthy, Evan J Livesey, Conceptualization, Investigation, Methodology, Writing—review and editing; Tyler Davis, Conceptualization, Resources, Data curation, Software, Formal analysis, Supervision, Funding acquisition, Validation, Investigation, Visualization, Methodology, Writing—original draft, Project administration, Writing—review and editing

### Author ORCIDs

Sean R O'Bryan [iD] http://orcid.org/0000-0003-0562-8211

### Ethics

Human subjects: Subjects provided written informed consent before taking part in the study, and all procedures involving human subjects were approved by the Texas Tech University Institutional Review Board.

### Decision letter and Author response

Decision letter https://doi.org/10.7554/eLife.36395.019
Author response https://doi.org/10.7554/eLife.36395.020

## Additional files

### Supplementary files

• Supplementary file 1. Number of trials where common and rare responses were made for each participant over the 24 ambiguous scene-object test trials.
DOI: https://doi.org/10.7554/eLife.36395.012

• Transparent reporting form
DOI: https://doi.org/10.7554/eLife.36395.013

### Data availability

Source data and scripts used to create all figures and tables (e.g. R code, PyMVPA scripts, statistical maps for the model-based fMRI analysis) are posted to a publicly available online repository (Open Science Framework: https://osf.io/atbz7/). Raw fMRI data for the study organized according to Brain Imaging Data Structure (BIDS) guidelines are available at https://openneuro.org/datasets/ds001302.

The following datasets were generated:

| Author(s) | Year | Dataset title | Dataset URL | Database, license, and accessibility information |
|---|---|---|---|---|
| Sean R O'Bryan | 2018 | Model-based fMRI reveals dissimilarity processes underlying base rate neglect | https://osf.io/atbz7/ | Publicly available at the Open Science Framework |
| Sean R O'Bryan, Darrell A Worthy, Evan J Livesey, Tyler Davis | 2018 | Inverse Base Rate | https://openneuro.org/datasets/ds001302 | Publicly available at the Open Neuro website (accession no. ds001302) |

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
