## [Decision Letter]

Thank you for submitting your article "Model-based fMRI reveals dissimilarity processes underlying base rate neglect" for consideration by *eLife*. Your article has been reviewed by three peer reviewers, including Timothy Verstynen as the Reviewing Editor and Reviewer #1, and the evaluation has been overseen by Michael Frank as the Senior Editor. The following individual involved in review of your submission has agreed to reveal their identity: Carol Seger (Reviewer #3).

The reviewers have discussed the reviews with one another and the Reviewing Editor has drafted this decision to help you prepare a revised submission.

Summary:

The reviewers all agreed that this was an important and creative study. The use of MVPA to test explicit psychological models of a cognitive phenomenon was innovative. However, all of the reviewers found the presentation of the theoretical and methodological aspects of the study to be opaque. This resulted in substantial difficulty in being able to accurately judge whether the key findings were veridical or not (or in some cases what the key findings are). Given the overall positive opinion on the scope and aims of the study, the article will require substantial revisions in order to be able to adequately evaluate the conclusions and interpretations put forth.

Below is a consolidated review of the comments from all reviewers. The major points are organized by topic.

Essential revisions:

1) Inferential concerns:

The authors are often imprecise when writing about the cognitive functions that might underlie the differences in neural activation in the MVPA analyses. They often refer to the subjects "thinking about" the common or rare category, but this is unclear. The subjects could consciously be thinking about the category. The representations could be activated implicitly, and indirectly and unconsciously bias the subjects' responses. The subjects could simply be paying more attention to the certain features, and this attention could precede (attending to a feature associated with a rare disease could bias the decision) or even follow the decision (deciding it is the rare disease heightens attention to features of the rare disease). The authors do briefly discuss alternatives in the Discussion section, but a more systematic analysis of the different options and which are ruled out by the present results versus which are still open questions would be helpful.

2) Ambiguity in the MVPA models:

What parameters from the dissGCM model were used?

It is unclear how the pattern similarity measures during trials when the low base rate outcome is selected is consistent with the authors' theory. However, it seems that the increased similarity to rare features on trials when the high base rate outcome is selected (for objects only) would be counter to the authors' dissGCM model.

Finally, the authors have a clear model-based definition of similarity and dissimilarity in the Model description section, similarity as the summed similarity to the winning category exemplars, and dissimilarity as the summed similarity to other categories, so that the two measures are relative to different sets of stimuli. They didn't clearly establish this difference in the Introduction and didn't always clearly interpret results clearly. Many readers will come to the paper assuming that dissimilarity is just the opposite of similarity and need to clearly understand when and how each term is being used. Perhaps using terms like Dissimilarity-Other Category and Similarity-Same category would make it clearer. Some of the most exciting results of the paper are the different neural correlates of similarity and dissimilarity (Figure 5), but it is easy to get confused as to why the negative similarity regressor (blue in Figure 5) differs from the dissimilarity regressor.

3) Logical Flow:

The authors rely heavily on assuming that the reader will not read the paper in a linear order, instead assuming the reader will be familiar with much of the Materials and methods when reading the Results. For example, information regarding the regions that the representational distance scores are drawn from is buried at the end of the Materials and methods. In addition, there are not clear details in the Materials and methods regarding the calculation of the similarity score itself. Thus, it is almost impossible for the reader to know how to interpret the key findings (Figures 3-5) without going back and forth or making some assumptions about the approach. This needs to be overhauled given the format of *eLife*.

The one place this isn't a concern is the presentation of the dissGCM model. While the details are appreciated, they are also out of place. This needs to be in the Results and Materials and methods, not the Introduction, and disambiguated so that conceptual information necessary for understanding the model and its fit to behavior is provided in Results, while the model itself is formally defined in the Materials and methods.

4) Neural Similarity analysis:

In many cases, it's not clear that the correct statistics were used. For example, in Figure 3, the interaction and main effects are reported as post-hoc t-tests. How was the interaction actually calculated (i.e., can't assess an interaction with a t-test)? What linear model was used? There is also a concern that the key analysis reported in Figure 3 may be based on very few trials. How many trials were actually included for each subject? Was there sufficient statistical power for the analysis?

5) Parameterization of behavior for brain-behavior correlations:

It is not immediately clear what the value is in showing the brain-behavior predictions based on the probability of selecting the low base rate outcome and the pattern similarity scores in Figure 5. This needs to be motivated and explained in more detail.

There is a specific model parameter, (*s* in Equation 3), that defines the degree to which a subject relies on similarity vs. dissimilarity. Wouldn't correlating individual differences in pattern similarities with this variable be a much more direct measure of the dissGCM hypothesis?

6) Value and clarity of univariate BOLD analysis:

The description of how these results are obtained is a bit opaque in the Materials and methods.

This analysis is asking a tangential question than the primary hypothesis of the paper (i.e., it's not supporting or adding nuance to the key findings from the multivariate analysis).

The interpretations from these maps is subjective and somewhat vague.

7) Analysis of face regions with face stimuli:

Why didn't the authors look at face activation in a face region (e.g., FFA) at all? Faces are the imperfect predictor here, which means they are the stimulus with the most flexibility in terms of associating with the outcome. Wouldn't a key prediction of the dissGCM model be that faces be more similar to object and scene area activity on trials when the common outcome is selected and vice versa for trials where the rare outcome is selected? It seems odd that the analysis is restricted to objects and scenes given the unique position face stimuli are given in the task.

8) Behavioral analysis:

For the behavioral analysis (Figure 2), was the interaction formally decomposed? Please report the relevant statistics in detail. It is also puzzling why reaction time was not reported. Reaction time could shed light on the mechanisms underlying choices. For example, if replying based on similarity is due to habitual responding, one would predict faster RTs in that case. One would also expect longer RTs for more deliberative choices.

9) Presentation of consecutive rare stimuli:

Are there any trials where the authors present two consecutive rare stimuli? Carvalho and Goldstone (2015) suggest that participants make decision by comparing the stimulus at time t with the stimulus at time t-1. In the current article, the stimulus at time t-1 is more likely to be from the same category as the stimulus at time t for common categories than for rare categories. Same category stimuli focus on within-category information (similarity) while different category stimuli focus on between-category information (dissimilarity). This alternative explanation does not rely on base rate and should be discussed.

---

## [Author Response]

Essential revisions1) Inferential concerns:The authors are often imprecise when writing about the cognitive functions that might underlie the differences in neural activation in the MVPA analyses. They often refer to the subjects "thinking about" the common or rare category, but this is unclear. The subjects could consciously be thinking about the category. The representations could be activated implicitly, and indirectly and unconsciously bias the subjects' responses. The subjects could simply be paying more attention to the certain features, and this attention could precede (attending to a feature associated with a rare disease could bias the decision) or even follow the decision (deciding it is the rare disease heightens attention to features of the rare disease). The authors do briefly discuss alternatives in the Discussion section, but a more systematic analysis of the different options and which are ruled out by the present results versus which are still open questions would be helpful.

Throughout the paper, we no longer refer to subjects thinking about the categories, and instead use more precise terms relating to what the MVPA analysis is objectively measuring (e.g., participants are activating patterns associated with the common category exemplar).

We have also revised the manuscript to more thoroughly discuss the cognitive processes that we believe the pattern similarity measure is capturing, and justify these predictions based on prior research using similar methods in concert with the representational assumptions made by both similarity- and dissimilarity-based theories of the IBRE. For instance, in the Introduction:

“In this context, we anticipate that the multivoxel pattern analysis will index an interactive process between feature-based attention and memory retrieval: both the dissGCM, pure similarity-based GCM, and previous dissimilarity-based inference models predict that categorization decisions are driven by an attention-weighted memory process whereby a stimulus is compared to the contents of memory. […] Because the dissGCM predicts greater contributions from the common, unchosen category during this retrieval process, we expect multivoxel patterns during ambiguous trials to reveal greater activation of information associated with the common, unchosen category.”

We then more thoroughly address the types of representations that may be contributing to our findings in the Discussion:

“We predict that our multivoxel results are not driven directly by simple feature-based attention, but instead indicate some combination of attention and memory-based retrieval of the category exemplars. […] Future studies may wish to combine multivoxel pattern analysis with eye-tracking (e.g., Leong et al., 2017) to better understand the unique contributions that attention and memory make to the present results.”

2) Ambiguity in the MVPA models:What parameters from the dissGCM model were used?

For both the multivoxel results and the model-based univariate results, our focus is on the predicted similarity- and dissimilarity-based contributions to the overall decision evidence *v*. We now thoroughly describe how the parameters for the multivoxel analysis (and the model-based imaging analysis) are obtained where appropriate in the Results text. For example:

“The prediction that information associated with the common category should be more active on ambiguous trials is derived from the dissGCM. […] For example, using these formulas, in the fitted version of the model, the proportion of the overall evidence for rare contributed by similarity to the rare category exemplar was nearly half the evidence contributed by dissimilarity to the common category exemplar (rare = 0.088; common = 0.153, in the dissGCM’s attention weighted similarity units).”

It is unclear how the pattern similarity measures during trials when the low base rate outcome is selected is consistent with the authors' theory. However, it seems that the increased similarity to rare features on trials when the high base rate outcome is selected (for objects only) would be counter to the authors' dissGCM model.

Greater neural similarity to the common features is consistent with the theory that people more heavily rely on (dissimilarity to) the common category to infer that the ambiguous cases must be rare, consistent with dissGCM’s predictions for the contribution of dissimilarity-based evidence to common category exemplars relative to similarity-based evidence for the rare exemplars (see above). We now clarify this result and offer a more thorough interpretation following the details of the statistical test in the Results section:

“To summarize, our multivoxel findings for the test phase suggest that people more strongly activate information associated with common categories when engaging in base rate neglect, consistent with dissGCM’s prediction that dissimilarity to the common category exemplar contributes more to rare decisions than similarity to the rare category exemplar. […] Our behavioral findings provide additional evidence for such a dissociation, as reaction times were found to be significantly slower for rare relative to common responses.”

Regarding the reviewers’ second point, we found that the difference between representational similarity to the rare and common objects when subjects chose the common category was not statistically significant, *t* (16) =. 869, p =. 398. Figure 3 is now collapsed across visual stimulus categories to more clearly illustrate the interaction effect between activation of rare/common stimulus features and whether the rare/common category was chosen. We believe that presenting the data in this manner will be more straightforward for readers to interpret, given that we now display the means for imperfect face predictors in the graph (see comment #6) and that no interactions or main effects of whether the stimulus was a scene or object were observed across cells. In addition to Figure 3 which is collapsed across objects/scenes, we now include a figure supplement (Figure 3—figure supplement 1) which depicts the means associated with each distinct stimulus (e.g., rare object) in the model. Increased neural similarity to the rare object category for common responses would indeed run counter to the predictions of dissGCM – as we now discuss in this section (see the paragraph above), dissGCM does not suggest an alternative process that leads to common responses, and instead is designed to account for participants’ overall tendency to choose the rare category in ambiguous cases.

Finally, the authors have a clear model-based definition of similarity and dissimilarity in the Model description section, similarity as the summed similarity to the winning category exemplars, and dissimilarity as the summed similarity to other categories, so that the two measures are relative to different sets of stimuli. They didn't clearly establish this difference in the Introduction and didn't always clearly interpret results clearly. Many readers will come to the paper assuming that dissimilarity is just the opposite of similarity and need to clearly understand when and how each term is being used. Perhaps using terms like Dissimilarity-Other Category and Similarity-Same category would make it clearer. Some of the most exciting results of the paper are the different neural correlates of similarity and dissimilarity (Figure 5), but it is easy to get confused as to why the negative similarity regressor (blue in Figure 5) differs from the dissimilarity regressor.

We agree that more clarity in our references to similarity/negative similarity vs. dissimilarity (and establishing what each represents in more detail) was necessary. The revised manuscript now sets up this distinction in the Introduction. For example:

“In addition to our multivoxel analysis, we also test whether using dissimilarity-based evidence against unchosen categories may tap distinct brain regions, such as the rlPFC, beyond those involved with similarity-based computations. […] We anticipated that the MTL and vmPFC would be positively associated with similarity-based evidence, whereas dlPFC would be negatively associated with similarity-based evidence for the winning category. Contrastingly, we expected rlPFC to track estimates of dissimilarity-based evidence against alternative options.”

Beyond introducing the point that similarity- and dissimilarity-based evidence are taken relative to the chosen and unchosen categories early in the paper, we now remind readers of this point in several relevant sections of the manuscript. In particular, we emphasize this distinction in the model-based univariate Results section where it is critical to differentiate between regions associated with dissimilarity-based evidence versus those negatively associated with similarity-based evidence. For example:

“To test whether similarity- and dissimilarity-based evidence rely on different brain regions, we modeled univariate voxel-wise activation using trial-by-trial estimates of similarity- and dissimilarity-based evidence derived from the dissGCM. […] The regions associated with dissimilarity-based evidence in this analysis are thus distinct from those negatively associated with similarity-based evidence because they are derived from evidence against the alternative, non-winning category.”

3) Logical Flow:The authors rely heavily on assuming that the reader will not read the paper in a linear order, instead assuming the reader will be familiar with much of the Materials and methods when reading the Results. For example, information regarding the regions that the representational distance scores are drawn from is buried at the end of the Materials and methods. In addition, there are not clear details in the Materials and methods regarding the calculation of the similarity score itself. Thus, it is almost impossible for the reader to know how to interpret the key findings (Figures 3-5) without going back and forth or making some assumptions about the approach. This needs to be overhauled given the format of eLife.

We now include detailed information about our multivoxel analysis in the Results section, including a description of how the ROIs were selected. Along with these details, we describe how the RSA measure was found to successfully differentiate control items during the test phase:

“For this analysis, multivoxel pattern estimates were anatomically restricted to ROIs in ventral temporal cortex associated with each visual stimulus category (objects: left inferior posterior temporal gyrus; scenes: bilateral parahippocampal gyrus; and faces: right temporal occipital fusiform gyrus). […] Likewise, estimates for faces on face-only control trials were found to be significantly greater than on object-only trials, t (21) = 2.54, p =. 019, and scene-only trials, t (21) = 2.92, p =. 008.”

We believe that in the previous version of the manuscript, it was not always clear whether we were referring to the pattern similarity measure or model-based predictions of similarity-based evidence, which may have been a point of confusion. The revised manuscript is now more precise with these terms by clearly establishing when we are referring to model-based predictions of similarity-based evidence to a given category or exemplar, versus BOLD pattern similarity in the multivoxel analysis.

The multivoxel pattern analysis section in the Materials and methods now follows a more logical organization, and additional details about the BOLD pattern similarity calculations were added:

“RSA was conducted using the PyMVPA toolbox (Hanke et al., 2009) and custom Python routines. […] For interpretative ease, the distances were converted to similarities using exp(- distance), and then standardized (z-scored) within participants.”

The one place this isn't a concern is the presentation of the dissGCM model. While the details are appreciated, they are also out of place. This needs to be in the Results and Materials and methods, not the Introduction, and disambiguated so that conceptual information necessary for understanding the model and its fit to behavior is provided in Results, while the model itself is formally defined in the Materials and methods.

We now include the formal model definition in the Materials and methods section, while including relevant conceptual information and details regarding the key model parameters where appropriate in the Results.

4) Neural Similarity analysis:In many cases, it's not clear that the correct statistics were used. For example, in Figure 3, the interaction and main effects are reported as post-hoc t-tests. How was the interaction actually calculated (i.e., can't assess an interaction with a t-test)? What linear model was used? There is also a concern that concerned that the key analysis reported in Figure 3 may be based on very few trials. How many trials were actually included for each subject? Was there sufficient statistical power for the analysis?

The multivoxel results for the test phase (Figure 3) were calculated using a linear mixed effects model with random intercepts for each participant. The interaction was previously reported as a *t*-value corresponding to the γ-weight for the interaction term, which by default are accompanied by a *t*-statistic in R’s “nlme” package. However, we agree that this may be a point of confusion for readers and as such now report the equivalent *F-*testfor the interaction:

“Multivoxel pattern similarity results for the ambiguous test trials are depicted in Figure 3. […] Interestingly, there was no significant difference between pattern similarity for rare and common features when participants made a common response, t (21) = 0.45, p =. 653.”

Based on this comment, we have also made it clear throughout the manuscript where linear mixed effects modeling was used and report the *F* statistic associated with the resulting interaction terms where applicable.

Regarding the statistical power of the results displayed in Figure 3, we anticipate that there was sufficient power for the test. The number of ambiguous test trials was doubled relative to the other trial types during the test phase with this analysis in mind, and each participant was given 24 ambiguous scene-object pairings to categorize over three scanning runs. In the Materials and methods, we now detail the number of trials allocated to each test stimulus, and a new supplementary file (Supplementary File 1) depicts the number of rare and common responses made on the key scene-object trials for each subject. Moreover, we now show that the pattern similarity measure accurately discriminates between faces, objects, and scenes within each functional localizer ROI for control (face-, object-, or scene-only) trials in the Results section, which we believe further alleviates concerns about the power of the analysis.

5) Parameterization of behavior for brain-behavior correlations:It is not immediately clear what the value is in showing the brain-behavior predictions based on the probability of selecting the low base rate outcome and the pattern similarity scores in Figure 5. This needs to be motivated and explained in more detail.

We now thoroughly discuss the motivation for this analysis in terms of its relationship to previous similarity- versus dissimilarity-based explanations of IBRE behavior (Results section):

“As with the test phase, dissimilarity-based theories make the somewhat counterintuitive prediction that it is specifically what people learn about the common category that is driving later choices of the rare category. […] This prediction is in direct contrast to the dominant similarity-based explanation of IBRE, which posits that it is specifically a stronger learned association between rare perfect predictors and their category that drives later rare category selections for the ambiguous test probes (Medin and Edelson, 1988; Kruschke, 1996, 2001).”

There is a specific model parameter, (s in Equation 3), that defines the degree to which a subject relies on similarity vs. dissimilarity. Wouldn't correlating individual differences in pattern similarities with this variable be a much more direct measure of the dissGCM hypothesis?

The dissGCM, like the GCM, is made to be a model of asymptotic categorization performance and is not well-equipped to capture individual differences. One reason for this is that the model is constrained to fit all trials, not just the key ambiguous trials. Similarity is critical for responses on many trials, and highly weighting dissimilarity leads the *s* parameter to be highly skewed in fits to individual subject data. Further, because there are three times more common items, the raw dissimilarity-based evidence is much higher for the common category than the similarity-based evidence for rare, and thus almost any amount of dissimilarity weighting produces the predicted rare responding. Thus, there is little pressure on the model to modulate the weighting of dissimilarity to fit the IBRE trials, and the model is very underidentified if fit to only the key IBRE trials.

We did fit an individual subject version nonetheless, finding a correlation between the weight to dissimilarity and slowed RTs. This may suggest that more deliberation is required to process dissimilarity, consistent with our MVPA results and the observed association between rlPFC and dissimilarity-based contributions to choice. No other relevant correlations were observed with the *s* parameter, but this is likely to due to the issues described above as its values were highly skewed when fit to individual subjects.

Importantly, because the *s* parameter is a scalar multiple of the similarity- and dissimilarity-based evidence, our model-based predictions for the univariate analysis are not affected by the *s* parameter value. For these reasons, we have decided not to include this analysis in the present study, but would be willing to do so if it was deemed critical by reviewers. We have included a discussion paragraph addressing this subject:

“The dissGCM, like the standard GCM, is a model of asymptotic categorization performance and generalization, and thus is not well-equipped to account for learning dynamics or individual differences. […] However, such a model would require extensive additional data to validate, and thus is beyond the scope of the present study.”

6) Value and clarity of univariate BOLD analysis:The description of how these results are obtained is a bit opaque in the Materials and methods.

We now detail the calculation of the trial level similarity- and dissimilarity-based evidence estimates and how they were used as regressors in the model-based univariate section of the Materials and methods:

“The first-level model included an EV for stimulus presentation and two model-based parametric modulators: Similarity- and dissimilarity-based evidence, computed from the dissGCM. Specifically, these regressors were obtained on a trial-by trial basis using Equation 3 (see Model section), where the evidence contribution of summed similarity to the winning category (C_A_; Most probable category according to the model) is calculated as:

s∑Sj∈CAtjsimij,(5)

and the evidence contribution of summed dissimilarity to non-winning categories with a positive feature match is calculated as:

1-s∑Sj∈¬CAtj(1-simij)(6)

Both parametric modulators were centered and scaled (z-scored) within run.”

This analysis is asking a tangential question than the primary hypothesis of the paper (i.e., it's not supporting or adding nuance to the key findings from the multivariate analysis).

We believe that the model-based imaging analysis provides critical support for the overall conclusions of the paper in that dissimilarity-based evidence not only contributes to base rate neglect, but that this type of processing is cognitively and neurobiologically distinct from pure similarity-based processing. While the neurobiological systems involved in similarity-based categorization are well understood, this is the first study to investigate whether incorporating contrastive evidence into categorization decisions recruits brain regions beyond those typically correlated with evidence in rule-based category learning (e.g., vmPFC and dlPFC). In the revised manuscript, we more thoroughly explain the rationale for this analysis in the Introduction. For example:

“It is possible that dissimilarity processes require manipulating similarity relationships between category representations in a more symbolic or abstract manner, as anticipated by previous dissimilarity theories. […] Specifically, we can test whether regions that are known to be critical for using higher-level abstract rules track dissGCM’s predicted trial-by-trial used of dissimilarity-based evidence and whether these regions diverge from those typically found to track estimates of similarity-based evidence.

Likewise:

“In the present study, dissimilarity-based generalization to novel feature pairings may depend on rule evaluation processes in the rlPFC more so than simple similarity-based processing, if studies anticipating that dissimilarity-based processes depend more upon higher-level symbolic rules are correct (Juslin, Wennerholm and Winman, 2001; Winman et al., 2005). Alternatively, pure similarity-based accounts suggest that generalization patterns in an IBRE task do not depend on the existence of a separate, higher-level mechanism (Medin and Edelson, 1988; Kruschke, 1996, 2001), and would thus expect a single neurobiological network associated with similarity-based processing to be engaged for choice across trials.”

We also re-emphasize these points when reporting our model-based univariate findings in the Results:

“Comparing the statistical maps in Figure 6A and 6B, it is apparent that greater relative contributions of dissimilarity-based evidence do not necessitate smaller contributions of similarity-based evidence in the dissGCM: if these regressors were anticorrelated, one would expect the regions associated with dissimilarity processes to resemble the fronto-parietal network we found to be negatively associated with similarity. Instead, our results show that using contrastive evidence uniquely engages the rlPFC, in line with dual-process theories that suggest dissimilarity-based processing may distinctly depend upon higher-level, abstract reasoning.”

The interpretations from these maps is subjective and somewhat vague.

We agree with the reviewers that the paper would benefit from making more connections between our univariate findings and the broader literature on categorization, rather than focusing somewhat exclusively on our a priori regions of interest. In line with this comment, we now present a considerably more detailed univariate Results section with paragraphs dedicated to each of the three contrasts. For example, where we discuss the positive correlates of similarity:

“Our analysis showed that greater similarity-based contributions to the winning category choice were associated with activation in the MTL (left hippocampus) vmPFC, and primary motor cortex (Figure 6A, depicted in red). […] While more anterior motor-planning regions such as pre-SMA and SMA tend to be associated with rule acquisition processes (e.g., Boettiger and D’Esposito, 2005), primary motor cortex has been found to track increasing levels of response automaticity in categorization tasks (Walschmidt and Ashby, 2011).”

7) Analysis of face regions with face stimuli:Why didn't the authors look at face activation in a face region (e.g., FFA) at all? Faces are the imperfect predictor here, which means they are the stimulus with the most flexibility in terms of associating with the outcome. Wouldn't a key prediction of the dissGCM model be that faces be more similar to object and scene area activity on trials when the common outcome is selected and vice versa for trials where the rare outcome is selected? It seems odd that the analysis is restricted to objects and scenes given the unique position face stimuli are given in the task.

Consistent with the reviewers’ suggestion, we analyzed BOLD pattern similarity to faces using the same ROI-selection technique as was used for objects and scenes (6 mm spheres around subject-specific activation peaks within right temporal occipital fusiform gyrus). For the test analysis, we now report and depict mean pattern similarity to faces for the ambiguous test trials alongside the object/scene patterns in Figure 3 and Figure 3—figure supplement 1. Pattern similarity to faces was not found to differ between trials where a common or rare response was made. We did not include the face dimension in the primary mixed effects model, with the following rationale now included in the manuscript:

“Because faces were off-screen for the key test trials and pattern similarity to the face dimension could represent information associated with either common or rare exemplars, no a priori predictions were made regarding pattern similarity to faces on ambiguous trials. However, a one-sample t-test revealed no significant differences in pattern similarity to the face dimension across responses, t (21) = 0.22, p =. 828. Mean pattern similarity to faces on the ambiguous test trials is depicted by the grey squares in Figure 3, with error bars absent to indicate that these values did not contribute to the main statistical model.”

Additionally, we now analyze pattern similarity to the imperfectly predictive faces during the learning phase in relation to subjects’ IBRE tendencies, as was previously done for the perfect common and rare predictors. Unlike the ambiguous test trials, faces were an on-screen stimulus dimension throughout the learning phase, and accordingly, serve as excellent controls for the perfect predictor correlations in this analysis. We now depict the respective relationships between pattern similarity to faces on common/rare disease trials and test behavior in Figure 5. The results of the brain-behavior correlations are also detailed in the text:

“Figure 5 depicts the associations between BOLD pattern similarity to common and rare stimulus dimensions during learning and base rate neglect. […] Likewise, the difference in positive slope between pattern similarity and base rate neglect for common versus rare perfect predictors was marginally significant, t (60) = 1.88, p =. 065.”

Please note that for the results above, the correlations for pattern similarity to the common and rare features have changed slightly. This is because the previous correlations were calculated using rare response probabilities obtained from our MVPA output files, which discard the final trial of each scanning run to ensure stable estimates. We now correctly include the final trial from the behavioral data in this analysis. The inclusion (or previous omission) of this final trial did not affect the interpretation or statistical significance of the previously reported correlations compared to what we find here.

8) Behavioral analysis:

For the behavioral analysis (Figure 2), was the interaction formally decomposed? Please report the relevant statistics in detail. It is also puzzling why reaction time was not reported. Reaction time could shed light on the mechanisms underlying choices. For example, if replying based on similarity is due to habitual responding, one would predict faster RTs in that case. One would also expect longer RTs for more deliberative choices.

For the learning curves presented in Figure 2, we now decompose the interaction effect using a linear mixed effects model to appropriately account for the nesting structure in the data (trials within participants) and report where categorization accuracy was significantly higher for the common disease during the learning phase. Blocks where significant differences were observed are now indicated in the figure as well.

We appreciate the reviewers’ suggestion to examine reaction time patterns in the data. Consistent with the idea that rare responses may involve a more cognitively demanding (dissimilarity-based) process, analyzing RTs for the different responses on ambiguous test trials revealed that rare responses were indeed made more slowly than common responses. These findings are now detailed in the behavioral Results section:

“In addition to response probabilities, we tested whether reaction times differed on the ambiguous test trials depending on whether a rare or common response was made. On these trials of interest, a linear mixed effects model revealed that RTs were considerably slower when participants made rare responses (M = 1.47 s) in comparison to common responses (M = 1.27 s), t (21) = 10.48, p <.001. The observation of slowed RTs on ambiguous trials receiving rare responses suggests that rare selections may be more cognitively demanding relative to common selections, consistent with previous dissimilarity-based theories of IBRE that posit a role of higher-level, inferential reasoning in base rate neglect.”

9) Presentation of consecutive rare stimuli:Are there any trials where the authors present two consecutive rare stimuli? Carvalho and Goldstone (2015) suggest that participants make decision by comparing the stimulus at time t with the stimulus at time t-1. In the current article, the stimulus at time t-1 is more likely to be from the same category as the stimulus at time t for common categories than for rare categories. Same category stimuli focus on within-category information (similarity) while different category stimuli focus on between-category information (dissimilarity). This alternative explanation does not rely on base rate and should be discussed.

We now include a Discussion section explaining how interleaved versus blocked designs (and how they create runs of same- vs. different-category trials) lead to differences in selective attention that emphasize features that are characteristic of a category (blocked, emphasizing within category) versus distinguishing between category differences (interleaved), and connect these findings to the interpretation of the present study. For example:

“Early studies on the role of order and blocking on IBRE are similar to current work on blocked versus interleaved learning in the broader categorization literature (Birnbaum et al., 2013; Carvalho and Goldstone, 2014, 2015; Goldwater et al., 2018). […] Contrastingly, our MVPA and univariate fMRI results show that pure similarity-based processing cannot fully explain the IBRE, and thus strongly suggest that dissimilarity processes contribute to the IBRE.”